# Vector Grimoire: Codebook-based Shape Generation under Raster Image Supervision

## Abstract

Scalable Vector Graphics (SVG) is a popular format on the web and in the design industry. However, despite the great strides made in generative modeling, SVG has remained underexplored due to the discrete and complex nature of such data. We introduce Grimoire, a text-guided SVG generative model that is comprised of two modules: A Visual Shape Quantizer (VSQ) learns to map raster images onto a discrete codebook by reconstructing them as vector shapes, and an Auto-Regressive Transformer (ART) models the joint probability distribution over shape tokens, positions, and textual descriptions, allowing us to generate vector graphics from natural language. Unlike existing models that require direct supervision from SVG data, Grimoire learns shape image patches using only raster image supervision which opens up vector generative modeling to significantly more data. We demonstrate the effectiveness of our method by fitting Grimoire for closed filled shapes on MNIST and for outline strokes on icon and font data, surpassing previous image-supervised methods in generative quality and the vector-supervised approach in flexibility.

## 1 Introduction

In the domain of computer graphics, Scalable Vector Graphics (SVG) has emerged as a versatile format, enabling the representation of 2D graphics with precision and scalability. SVG is an XML-based vector graphics format that describes a series of parametrized shape primitives rather than a limited-resolution raster of pixel values. While modern generative models have made significant advancements in producing high-quality raster images (Ho et al., 2020; Isola et al., 2017; Saharia et al., 2022; Nichol et al., 2021), SVG generation remains a less explored task. Existing works that have aimed to train a deep neural network for this goal primarily adopted language models to address the problem (Wu et al., 2023; Tang et al., 2024). In general, existing approaches share two key limitations: they necessitate SVG data for direct supervision which inherently limits the available data and increases the burden of data pre-processing, and they are not easily extendable when it comes to visual attributes such as color or stroke properties. The extensive pre-processing is required due to the diverse nature of an SVG file that can express shapes as a series of different basic

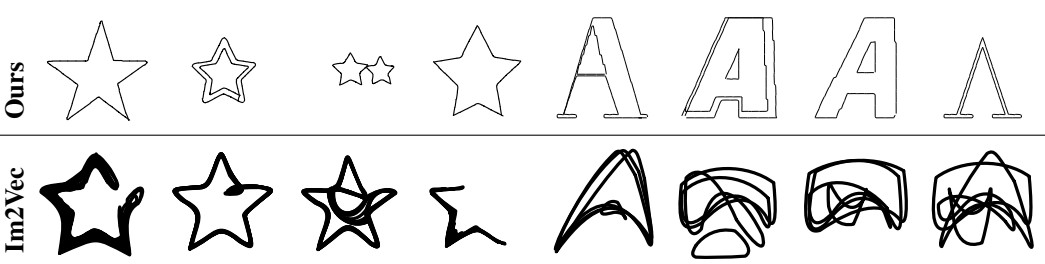

Figure 1: Generative results for fonts and icons from Grimoire and Im2Vec. Since Im2Vec does not accept any conditioning, we sample after training Im2Vec only on icons of stars or the letter A, respectively. For Grimoire we use the models trained on the full dataset conditioned on the respective class.

primitives such as circles, lines, and squares – each having different properties – that can overlap and occlude each other.

An ideal generative model for SVG should however benefit from *visual guidance for supervision*, which is not possible when merely training to reproduce tokenized SVG primitives, as there is no differentiable mapping to the generated raster imagery. In this paper, we present GRIMOIRE (Shape Generation with raster image supervision), a novel pipeline explicitly designed to generate SVG files with only raster image supervision. Our approach incorporates a differentiable rasterizer, DiffVG (Li et al., 2020), to bridge the vector graphics primitives and the raster image domain. We adopt a VQ-VAE recipe (Van Den Oord et al., 2017), which pairs a codebook-based discrete auto-encoder with an auto-regressive Transformer that models the image space implicitly by learning the distribution of codes that resemble them. We find this approach particularly promising for vector graphics generation, as it breaks the complexity of this task into two stages. In the first stage of our method, we decompose images into primitive shapes represented as patches. A vector-quantized auto-encoder learns to encode and map each patch into a discrete codebook, and decode these codes to an SVG approximation of the input patch, which is trained under raster supervision. In the second stage, the series of raster patches containing primitives are encoded and the prior distribution of codes is learned by an auto-regressive Transformer model conditioned on a textual description. At inference, a full series of codes can be generated from textual input, or other existing shape codes. Therefore, GRIMOIRE supports text-to-SVG generation and SVG auto-completion as possible downstream tasks out-of-the-box.

The key contributions of this work are:

1. We frame the problem of image-supervised SVG generation as the prediction of a series of individual shapes and their positions on a shared canvas.

2. We train the first text-conditioned generative model that learns to draw vector graphics with only raster image supervision.

3. We compare our model with alternative frameworks showing superior performance in generative capabilities on diverse datasets.

4. We release the code of this work to the research community[1].

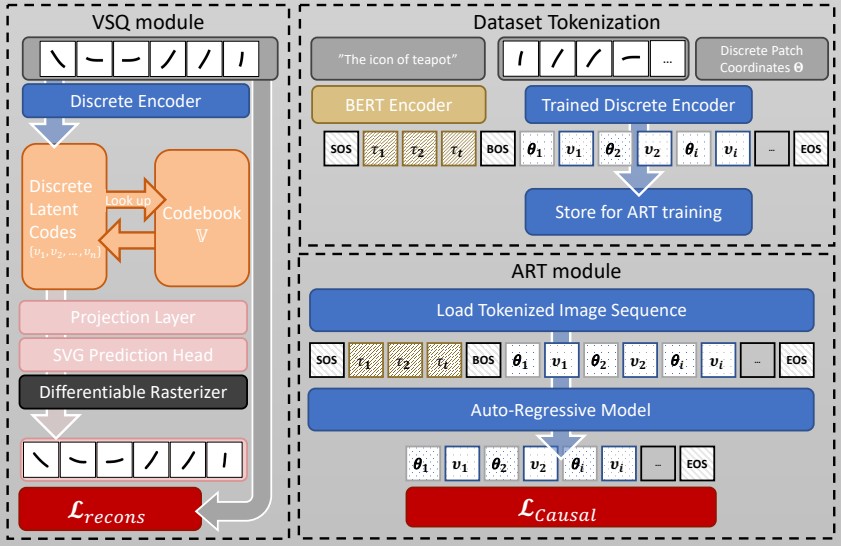

Figure 2: Overview of GRIMOIRE. On the left, the training process of our VSQ module is depicted, where raster input patches are encoded into discrete codes and reconstructed as SVG shapes using visual supervision. In the top right, each image is encoded into a series of discrete codes using the trained VSQ encoder and its textual description. The bottom right illustrates how the ART module learns the joint distribution of these codes and the corresponding text.

---

[1] https://github.com/under-review-papercode/9973

## 2 RELATED WORK

### 2.1 SVG GENERATIVE MODELS

The field of vector graphics generation has witnessed increasing interest. Following the extraordinary success of Large Language Models (LLM), the most recent approaches (Lopes et al., 2019; Aoki & Aizawa, 2022; Wu et al., 2023; Tang et al., 2024) have recast the problem as an NLP task, learning a distribution over tokenized SVG commands. Iconshop (Wu et al., 2023) introduced a method of tokenizing SVG paths that makes them suitable input for causal language modeling. To add conditioning, they employed a pre-trained language model to tokenize and embed textual descriptions, which are concatenated with the SVG tokens to form sequences that the auto-regressive Transformer can learn a joint probability on. StrokeNUWA (Tang et al., 2024) introduced Vector Quantized Strokes to compress SVG strokes into a codebook with SVG supervision and fine-tune a pre-trained Encoder–Decoder LLM to predict these tokens given textual input. However, both of these approaches suffer from a number of limitations. First, they require a corpus of SVG data for training, which hinges upon large pre-processing pipelines to remove redundancies, convert non-representable primitives, and standardize the representations. Secondly, there is no supervision of the visual rendering, which makes the models prone to data quality errors, e.g., excessive occlusion of shapes. Finally, these models lack any straightforward extensibility towards the inclusion of new visual features such as colours, stroke widths, or fillings and alpha values.

Hence, another line of work has sought to incorporate visual supervision. These approaches generally rely on recent advances in differentiable rasterization, which enables backpropagation of raster-based losses through different types of vectorial primitives such as Bézier curves, circles, and squares. The most important development in this area is DiffVG (Li et al., 2020), which removed the need for approximations and introduced techniques to handle antialiasing. They further pioneered image-supervised SVG generative models by training a Variational Autoencoder (VAE) and a Generative Adversarial Network (GAN) (Goodfellow et al., 2014) on MNIST (LeCun et al., 1998) and QuickDraw (Ha & Eck, 2017). These generative capabilities have subsequently been extended in Im2Vec (Reddy et al., 2021), which adopts a VAE including a recurrent neural network to generate vector graphics as sets of deformed and filled circular paths, which are differentiably composited and rasterized, allowing for back-propagation of a multi-resolution MSE-based pyramid loss. However, all of these models lack versatile conditioning (such as text) and focus on either image vectorization, i.e., the task of creating the closest vector representation of a raster prior, or vector graphics interpolation. We show in Section 5 that these approaches fail to capture the diversity and complexity of datasets such as FIGR-8, and generate repetitive samples.

A different type of SVG generation enabled by DiffVG is painterly rendering (Ganin et al., 2018; Nakano, 2019), where an algorithm iteratively fits a given set of vector primitives to match an image, guided by a deep perceptual loss function. To achieve this goal, CLIPDraw (Frans et al., 2022) rasterized a set of randomly initialized SVG paths and encoded these with a pre-trained CLIP (Radford et al., 2021) image encoder, iteratively minimizing the cosine distance between such embeddings and the text description. A similar approach was adopted by CLIPasso (Vinker et al., 2022) to translate images into strokes. Vector Fusion (Jain et al., 2023) leveraged Score Distillation Sampling (SDS) (Poole et al., 2022) to induce abstract semantic knowledge from an off-the-shelf Stable Diffusion model (Rombach et al., 2022).

### 2.2 VECTOR QUANTIZATION

VQ-VAE (Van Den Oord et al., 2017) is a well-known improved architecture for training Variational Autoencoders (Kingma & Welling, 2013; Rezende et al., 2014). Instead of focusing on representations with continuous features as in most prior work (Vincent et al., 2010; Denton et al., 2016; Hinton & Salakhutdinov, 2006; Chen et al., 2016), the encoder in a VQ-VAE emits discrete rather than continuous codes. Each code maps to the closest embedding in a codebook of limited size. The decoder learns to reconstruct the original input image from the chosen codebook embedding. Both the encoder–decoder architecture and the codebook are trained jointly. After training, the autoregressive distribution over the latent codes is learnt by a second model, which then allows for generating new images via ancestral sampling. Latent discrete representations were already pioneered in previous work (Mnih & Gregor, 2014; Courville et al., 2011), but none of the above methods close the performance gap of VAEs with continuous latent variables, where one can use the Gaussian

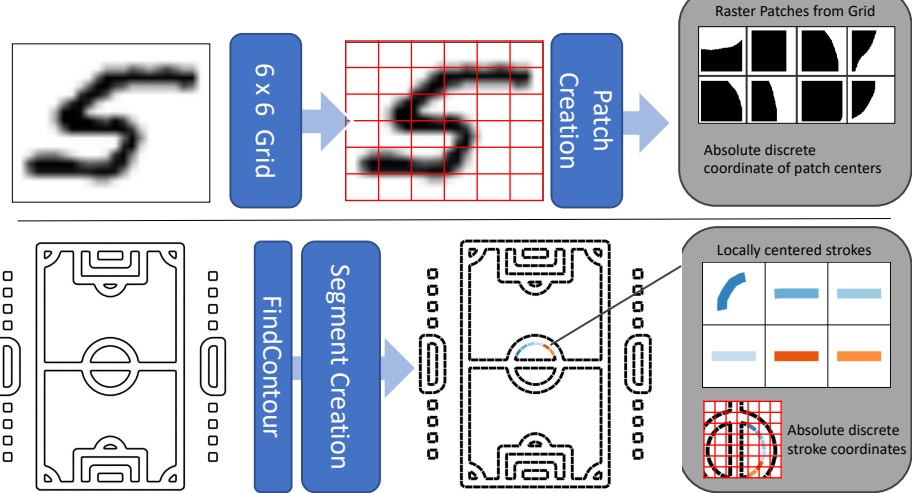

Figure 3: Overview of the data generation process for GRIMOIRE. For the MNIST digits, we simply create patches from a $6 \times 6$ Grid. For FIGR-8, we extract the outlines of each icon and create small centered raster segments. We save the original anchor position of each segment for the second stage of our training pipeline. More information about the outline extraction is provided in Section A.4. Fonts comes in vector format and can be easily manipulated to extract strokes, similarly to FIGR-8.

reparametrization trick, which benefits from much lower variance in the gradients. Mentzer et al. (2023) simplified the design of the vector quantization in VQ-VAE with a scheme called finite scalar quantization (FSQ), where the encoded representation of an image is projected to the nearest position on a low-dimensional hypercube. In this case, no additional codebook must be learned, but rather it is given implicitly, which simplifies the loss formulation. Our work builds in part on the VQ-VAE framework and includes the FSQ mechanism.

## 3 METHOD

### 3.1 STAGE 1 – VISUAL SHAPE QUANTIZER

The first stage of our model employs a **V**isual **S**hape **Q**uantizer (VSQ), a vector-quantized auto-encoder, whose encoder $E_{\text{VSQ}}$ maps an input image $I$ onto a discrete codebook $\mathbb{V}$ through vector-quantization and decodes that quantized vector into shape parameters of cubic Bézier curves through the decoder $D_{\text{VSQ}}$. Instead of learning the codebook (Van Den Oord et al., 2017), we adopt the more efficient approach of defining our codebook $V$ as a set of equidistant points in a hypercube with $q$ dimensions. Each dimension has $l$ unique values: $L = [l_1, l_2, \dots, l_q]$. The size of the codebook $|\mathbb{V}|$ is hence defined by the product of values of all $q$ dimensions. We define $q = 5$ and $L = [7, 5, 5, 5, 5]$ for a target codebook size of 4,375 unique codes, following the recommendations of the original authors (Mentzer et al., 2023).

Before being fed to the encoder $E_{\text{VSQ}}$, each image $I \in \mathbb{R}^{C \times H \times W}$ is divided into patches $\mathbf{S} = (s_1, s_2, \dots, s_n)$, with $s_i \in \mathbb{R}^{C \times 128 \times 128}$, where $C = 3$ is the number of channels. A set of discrete anchor coordinates $\mathbf{\Theta} = (\theta_1, \theta_2, \dots, \theta_n)$ with $\theta_i \in \mathbb{N}^2$ being the center coordinate of $s_i$ in the original image $I$ is also saved. The original image $I$ can then be reconstructed using $S$ and $\Theta$.

We experiment on three datasets (see Section 4). For MNIST, the patches are obtained by tiling each image into a $6 \times 6$ grid. For Fonts and FIGR-8, each patch depicts part of the target outline as shown in Figure 3.

The VSQ encoder $E_{\text{VSQ}}$ maps each patch $s_i \in \mathbb{R}^{C \times 128 \times 128}$ to $\xi$ codes on the hypercube $E_{\text{VSQ}}$ : $\mathbb{R}^{C \times 128 \times 128} \mapsto V$ as follows. Each centered raster patch $s_i$ is encoded with a ResNet-18 (He et al., 2016) into a latent variable $z_i \in \mathcal{Z} \subset \mathbb{R}^{d \times \xi}$ with $d = 512$. Successively, each of the $\xi$ codes is projected to $q$ dimensions through a linear mapping layer and finally quantized, resulting in $\hat{z}_i \in \mathbb{N}^q$.

The final code value $v_i \in \mathbb{V}$ is then computed as the weighted sum of all $q$ dimensions of $\hat{z}_i$:

$$v_i = \sum_{j=1}^{q} \hat{z}_{ij} \cdot b_j, \tag{1}$$

where the basis $b_j$ is derived as $b_j = \prod_{k=1}^{j-1} l_k$, with $b_1 = 1$. This transformation ensures that each unique combination of quantized values $\hat{z}_i$ is mapped to a unique code $v_i$ in the codebook $\mathbb{V}$.

This approach avoids auxiliary losses on the codebook while maintaining competitive expressiveness.

The decoder $D_{\text{VSQ}}$ consists of a projection layer, which transforms all the $\xi$ predicted codes back into the latent space $\mathcal{Z}$, and a lightweight neural network $\Phi_{\text{points}}$, which predicts the control points of $\nu$ cubic Bézier curves that form a single connected path.

Finally, the predicted path of $\nu$ Bézier curves from $\Phi_{\text{points}}$ passes through the differentiable rasterizer to obtain a raster output $\hat{s}_i = \text{DiffVG}(D_{\text{VSQ}}(E_{\text{VSQ}}(s_i)))$. In order to learn to reconstruct strokes and shapes, we train the VSQ module using the mean squared error:

$$\mathcal{L}_{\text{recons}} = (s - \hat{s})^2. \tag{2}$$

$D_{\text{VSQ}}$ can be extended to predict continuous values for any visual attribute supported by the differentiable rasterizer. Hence, we also propose series of other fully-connected prediction heads that can optionally be enabled: $\Phi_{\text{width}} : \mathcal{Z} \mapsto \mathbb{R}$ predicts the stroke width of the overall shape, and $\Phi_{\text{color}} : \mathcal{Z} \mapsto \mathbb{R}^{\mathbb{C}}$ outputs the stroke color or the filling color for the output of $\Phi_{\text{points}}$. All the modules are followed by a sigmoid activation function.

While $\mathcal{L}_{\text{recons}}$ would suffice for training the VSQ, operating only on the visual domain could lead to degenerate strokes and undesirable local minima. To mitigate this, we propose a novel geometric constraint $\mathcal{L}_{\text{geom}}$, which punishes control point placement of irregular distances measured between all combinations of points predicted by $\Phi_{\text{points}}$.

Let $P = (p_1, p_2, ..., p_{\nu+1})$ be the set of all start and end points of a stroke with $p_i = (p_i^x, p_i^y)$ and $p_i^x, p_i^y \in [0, 1]$. Then $\rho_{i,j}$ is defined as the Euclidean distance between two points $p_i$ and $p_j$, $\overline{\rho}_j$ is defined as the mean scaled inner distance for point $p_j$ to all other points in $P$, and $\delta_j$ as the average squared deviation from that mean for point $p_j$:

$$\overline{\rho}_j = \frac{1}{\nu} \sum_{\substack{i=1 \\ i \neq j}}^{\nu+1} \frac{\rho_{i,j}}{|i-j|} \qquad \delta_j = \frac{1}{\nu} \sum_{\substack{i=1 \\ i \neq j}}^{\nu+1} \left( \frac{\rho_{i,j}}{|i-j|} - \overline{\rho}_j \right)^2 \tag{3}$$

$\mathcal{L}_{\text{geom}}$ is finally defined as the average of the deviations for all start and end points in $P$. $\mathcal{L}_{\text{geom}}$ is then weighted with $\alpha$ and added to the reconstruction loss.

$$\mathcal{L}_{\text{geom}} = \frac{1}{\nu+1} \sum_{j=1}^{\nu+1} \delta_j \qquad \mathcal{L}_{\text{VSQ}} = \mathcal{L}_{\text{recons}} + \alpha \times \mathcal{L}_{\text{geom}} \tag{4}$$

With $\alpha$ being an hyper-parameter. The overall scheme of GRIMOIRE including the first stage of training is depicted in Figure 2.

## 3.2 STAGE 2 – AUTO-REGRESSIVE TRANSFORMER

After the VSQ is trained, each patch $s_i$ can be mapped onto an index code $v_i$ of the codebook $V$ using the encoder $E_{\text{VSQ}}$ and the quantization method. However, the predicted patch $\hat{s}_i$ captured by the VSQ does not describe a complete SVG, as the centering leads to a loss of information about their global position $\theta_i$ on the original canvas. Also, the sequence of tokens is still missing the text conditioning. This is addressed in the second stage of GRIMOIRE. The second stage consists of an **A**uto-**R**egressive **T**ransformer (ART) that learns for each image $I$ the joint distribution over the text, positions, and stroke tokens. A textual description $T$ of $I$ is tokenized into $\mathcal{T} = (\tau_1, \tau_2, \ldots, \tau_t)$ using a pre-trained BERT encoder (Devlin et al., 2018) and embedded. $I$ is visually encoded by transforming its patches $s_i$ onto $v_i \in V$ via the encoder $E_{\text{VSQ}}$, whereas each original patch position

$\theta_i \in \Theta$ is mapped into the closest position in a $256 \times 256$ grid resulting in $256^2$ possible position tokens. Special tokens `<SOS>`, `<BOS>`, and `<EOS>` indicate the start of a full sequence, beginning of the patch token sequence, and end of sequence, respectively. Each patch token is alternated with its position token. The final input sequence for a given image to the ART module becomes:

$$x = (\texttt{<SOS>}, \tau_1, \ldots, \tau_t, \texttt{<BOS>}, \theta_1, v_1, \ldots \theta_n, v_n, \texttt{<EOS>})$$

The total amount of representable token values then has a dimensionality of $|V| + 256^2 + 3 = 69,914$ for $|V| = 4{,}375$. A learnable weight matrix $W \in \mathbb{R}^{d \times 69,914}$ embeds the position and visual tokens into a vector of size $d$. The BERT text embeddings are projected into the same $d$-dimensional space using a trainable linear mapping layer. The ART module consists of 12 and 16 standard Transformer decoder blocks with causal multi-head attention with 8 attention heads for fonts and icons, respectively. The final loss for the ART module is defined as:

$$\mathcal{L}_{\text{Causal}} = -\sum_{i=1}^{N} \log p(x_i \mid x_{<i}; \theta) \tag{5}$$

During inference, the input to the ART module is represented as $x = (\texttt{<SOS>}, \tau_1, \ldots, \tau_t, \texttt{<BOS>})$, where new tokens are predicted auto-regressively until the `<EOS>` token is generated. Additionally, visual strokes can be incorporated into the input sequence to condition the generation process.

## 4 DATA AND EXPERIMENTAL SETTING

**MNIST.** We conduct our initial experiments on the MNIST dataset (LeCun et al., 1998). We upscale each digit to $128 \times 128$ pixels and generate the texual description using the prompt "*x* in black color", where *x* is the class of each digit. We adopt the original train and test split.

**Fonts.** For our experiments on fonts, we use a subset of the SVG-Fonts dataset (Lopes et al., 2019). We remove fonts where capital and lowercase glyphs are identical, and consider only 0–9, a–z, and A–Z glyphs, which leads to 32,961 unique fonts for a corpus of ∼2M samples. The font features – such as type of character or style – are extracted from the *.TTF* file metadata. The final textual description for a sample glyph $g$ in font style $s$ is built using the prompt: "[capital] $g$ in $s$ font", where "capital " is included only for the glyphs A-Z. We use 80%, 10%, and 10% for training, testing, and validation respectively.

**FIGR-8.** We validate our method on more complex data and further use a subset of FIGR-8 (Clouâtre & Demers, 2019), where we select the 75 majority classes (excluding "arrow") and any class that contains those, e.g., the selection of "house" further entails the inclusion of "dog house". This procedure yields 427K samples, of which we select 90% for training, 5% for validation, and 5% for testing. We use the class names as textual descriptions without further processing besides minor spelling correction. Since the black strokes of FIGR-8 mark the background rather than the actual icon, we invert the full dataset before applying our additional pre-processing described in Section A.4.

**Experimental Setup.** When training on FIGR-8, we utilize a contour-finding algorithm (Lorensen & Cline, 1987) to extract outlines from raster images, which are then divided into several shorter segments. Additional details regarding this extraction process can be found in Section A.4. In contrast, the Fonts dataset is natively available in vector format, making it easier to manipulate, similar to icons, before undergoing rasterization.

We propose two variants of $\Phi_{\text{points}}$ described in Section 3.1, a fully-connected neural network $\Phi_{\text{points}}^{\text{stroke}} : \mathcal{Z} \mapsto \mathbb{R}^{(2 \times (\nu \times 3 + 1))}$, which predicts connected strokes, and a 1-D CNN $\Phi_{\text{points}}^{\text{shape}} : \mathcal{Z} \mapsto \mathbb{R}^{(2 \times (\nu \times 3))}$, which outputs a closed shape.

We use $\mathcal{L}_{\text{geom}}$ only for the experiments with $\Phi_{\text{points}}^{\text{stroke}}$ and set $\alpha = 0.4$. We opt to train the ResNet encoder from scratch during this stage, since the target images belong to a very specific domain. The amount of trainable parameters is $15.36M$ for the encoder and $0.8M$ for the decoder. We stress the importance of the skewed balance between the two parameter counts, as the encoding of images is only required for training the model and encoding the training data for the auto-regressive Transformer in the next step. The final inference pipeline discards the encoder and only requires the trained decoder $D_{\text{VSQ}}$, hence resulting in more lightweight inference.

## 5    RESULTS

This section presents our findings in two primary categories.  First, we examine the **quality** of the reconstructions and generations produced by GRIMOIRE in comparison to existing methods. Second, we highlight the **flexibility** of our approach, demonstrating how GRIMOIRE can be easily extended to incorporate additional SVG features.

### 5.1    RECONSTRUCTIONS

**Closed Paths**. We begin by presenting the reconstruction results of our VSQ module on the MNIST dataset. In our experiments, we model each patch shape using a total of 15 segments. Increasing the number of segments beyond this point did not yield any significant improvement in reconstruction quality. Given the simplicity of the target shapes, we adopted a single code per shape.

We also conducted a comparative analysis of the reconstruction capabilities of our VSQ module against Im2Vec. To assess the generative quality of our samples, we employed the Fréchet Inception Distance (FID) (Heusel et al., 2017) and CLIPScore (Radford et al., 2021), both of which are computed using the image features of a pre-trained CLIP encoder. Additionally, to validate our VSQ module, we considered the reconstruction loss $\mathcal{L}_{recons}$, as it directly reflects the maximum achievable performance of the network and provides a more reliable metric.

As shown in Table 1, our VSQ module consistently achieves a lower reconstruction error compared to Im2Vec across all MNIST digits. In Table 2, we also report the reconstruction error for a subset of the dataset, selecting the digit zero due to its particularly challenging topology. Again, our method exhibits superior performance with lower reconstruction errors. For MNIST, we fill the predicted shapes from Im2Vec, since the raster ground truth images are only in a filled format. However, we present both filled and unfilled versions for all other scenarios.

The CLIPScore of our reconstructions is higher in both cases. Notably, FID is the only metric where Im2Vec occasionally shows superior results. We attribute this to the lower resolution of the ground truth images, which introduces instability in the FID metric. The CLIPScore, however, mitigates this issue by comparing the similarity with the textual description.

**Strokes**. For Fonts and FIGR-8, we conduct a deeper investigation to validate the reconstruction errors of VSQ under different configurations, varying the amount of segments and codes per shape, and the maximum length of the input strokes.  Our findings show that for Fonts, more than one segment per shape consistently degrades the reconstruction quality, possibly because the complexity of the strokes in our datasets does not require many Beziér curves to reconstruct an input patch. We also find that shorter thresholds on the stroke length help the reconstruction quality, as the MSE decreases when moving from 11% to 7% and eventually to 4% of the maximum stroke length with respect to the image size. Intuitively, shorter strokes are easier to model, but could also lead to very scattered predictions for overly short settings.

The best reconstructions are achieved by using multiple codes per centered stroke. The two-codes configuration has an average decrease in MSE of 18.28%, 41.46%, and 26.09% for the respective stroke lengths. However, the best-performing configuration with two codes per shape is just 11.36% better than the best single code representative, which we believe does not justify twice the number of required visual tokens for the second stage training. Throughout our experiments, the configurations

| Model | MNIST | | | Fonts | | | FIGR-8 | | |
|---|---|---|---|---|---|---|---|---|---|
| | MSE $\Downarrow$ | FID $\Downarrow$ | CLIP $\Uparrow$ | MSE $\Downarrow$ | FID $\Downarrow$ | CLIP $\Uparrow$ | MSE $\Downarrow$ | FID $\Downarrow$ | CLIP $\Uparrow$ |
| Im2Vec (filled) | 0.140 | **1.33** | 25.02 | 0.140 | 2.04 | 26.82 | 0.330 | 16.10 | 26.17 |
| Im2Vec | *n/a* | *n/a* | *n/a* | 0.050 | 5.64 | 26.72 | 0.050 | 13.90 | 26.17 |
| VSQ | **0.090** | 7.09 | **25.24** | 0.014 | 4.45 | 28.61 | 0.004 | 1.42 | 31.09 |
| VSQ + PI | *n/a* | *n/a* | *n/a* | **0.011** | **0.29** | **28.96** | **0.002** | **0.05** | **32.03** |

Table 1: Results for reconstructions of GRIMOIRE and Im2Vec on the test-set including all classes. The last row includes post-processing.

with multiple segments do consistently benefit from our geometric constraint. Ultimately, for our final experiments we choose ($\nu = 2, \xi = 1$) for Fonts, and ($\nu = 4, \xi = 2$) for FIGR-8.

Regarding the comparison with Im2Vec, Table 2 shows that the text-conditioned GRIMOIRE on a single glyph or icon has superior reconstruction performance even if Im2Vec is specifically trained on that subset of data. In Table 1, we also report the values after training on the full datasets. In this case, GRIMOIRE substantially outperforms Im2Vec, which is unable to cope with the complexity of the data.

Finally, as GRIMOIRE quickly learns to map basic strokes or shapes onto its finite codebook and due to the similarities between those primitive traits among various samples in the dataset, we find GRIMOIRE to converge even before completing a full epoch on any dataset. Despite the reconstruction error being considerably higher, we also notice reasonable domain transfer capabilities between FIGR-8 images and Fonts when training the VSQ module only on one dataset and keeping the maximum stroke length consistent. Qualitative examples of the re-usability of the VSQ module are reported in the Appendix.

| Model | MNIST (0) | | | Fonts (A) | | | Icons (Star) | | |
| | MSE ↓↓ | FID ↓↓ | CLIP ↑↑ | MSE ↓↓ | FID ↓↓ | CLIP ↑↑ | MSE ↓↓ | FID ↓↓ | CLIP ↑↑ |
|---|---|---|---|---|---|---|---|---|---|
| Im2Vec (filled) | 0.218 | **2.20** | 24.61 | 0.087 | 1.64 | 26.27 | 0.120 | 2.40 | 30.90 |
| Im2Vec | *n/a* | *n/a* | *n/a* | 0.060 | 6.33 | 25.78 | 0.110 | 11.17 | 30.40 |
| VSQ | **0.130** | 11.2 | **26.68** | 0.020 | 4.50 | 29.13 | 0.002 | 1.26 | 31.64 |
| VSQ + PI | *n/a* | *n/a* | *n/a* | **0.012** | **0.61** | **29.46** | **0.001** | **0.07** | **32.94** |

Table 2: Results for reconstructions of GRIMOIRE and Im2Vec on the test-set, using the class reported next to the dataset name. The last row includes post-processing.

## 5.2 GENERATIONS

**Text Conditioning.** We compare GRIMOIRE with Im2Vec by generating glyphs and icons and handwritten digits, and report the results in Table 3. Despite Im2Vec being tailored for single classes only, our general model shows superior performance in CLIPScore for all datasets. Im2Vec shows a generally lower FID score in the experiments with filled shapes, which we attribute again to the lower resolution of the ground truth images (MNIST) and a bias in the metric itself as CLIP struggles to produces meaningful visual embeddings for sparse images (Chowdhury et al., 2022) as for Fonts, FIGR-8. In contrast, in the generative results on unfilled shapes, GRIMOIRE almost consistently outperforms Im2Vec by a large margin for glyphs and icons.

Note that we establish new baseline results for the complete datasets, as Im2Vec does not support text or class conditioning.

Looking at qualitative samples in Figure 4 and Figure 1, one can see that contrary to the claim that surplus shapes collapse to a point (Reddy et al., 2021), there are multiple redundant shapes present in the generations of Im2Vec. A single star might then be represented by ten overlapping almost identical paths. The qualitative results in Figure 5 confirm this behaviour on the MNIST dataset. We also show that setting Im2Vec to predict only one single SVG path leads the model to compress the shape area and use its filling as a stroke width.

Overall, GRIMOIRE produces much cleaner samples with less redundancy, which makes them easier to edit and visually more pleasing. The text conditioning also allows for more flexibility. The

| Model | MNIST (0) | | MNIST (Full) | | Fonts (A) | | Fonts (Full) | | FIGR-8(Star) | | FIGR-8(Full) | |
| | FID ↓↓ | CLIP ↑↑ | FID ↓↓ | CLIP ↑↑ | FID ↓↓ | CLIP ↑↑ | FID ↓↓ | CLIP ↑↑ | FID ↓↓ | CLIP ↑↑ | FID ↓↓ | CLIP ↑↑ |
|---|---|---|---|---|---|---|---|---|---|---|---|---|
| Im2Vec (filled) | **2.22** | 24.69 | *n/a* | *n/a* | **1.20** | 25.81 | *n/a* | *n/a* | **2.97** | 31.72 | *n/a* | *n/a* |
| Im2Vec | *n/a* | 25.21 | *n/a* | *n/a* | 5.36 | 25.39 | *n/a* | *n/a* | 11.59 | 31.88 | *n/a* | *n/a* |
| GRIMOIRE (ours) | 12.25 | **26.60** | 9.25 | 25.25 | 5.61 | **30.60** | 1.67 | 28.64 | 6.25 | **32.24** | 0.64 | 29.00 |

Table 3: Results generations of GRIMOIRE and Im2Vec. GRIMOIRE is trained with all the classes of the dataset and conditioned to the respective class using the text description. FID uses test-data as a target.

generations are also diverse, as can be seen in Figure 4 where we showcase multiple generations for the same classes from FIGR-8. Additional generations on all datasets are provided in the Appendix.

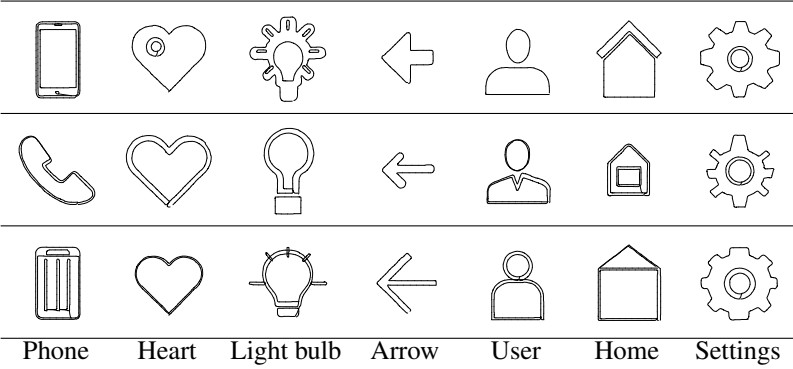

Phone    Heart    Light bulb    Arrow    User    Home    Settings

Figure 4: Examples of text-conditioned icon generation from GRIMOIRE.

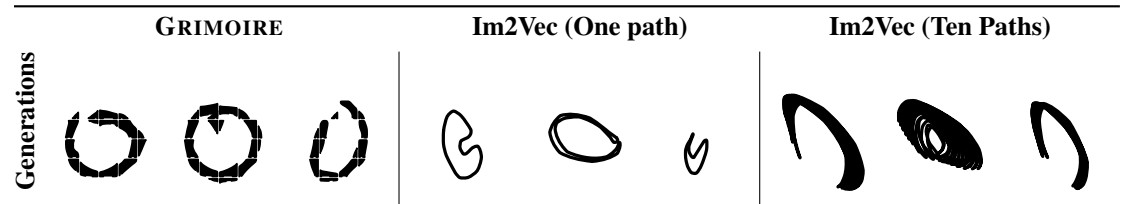

Figure 5: Generative results for the MNIST dataset from GRIMOIRE and Im2Vec with the number of predicted paths fixed to one and ten respectively. Since Im2Vec does not accept any conditioning, we sample after training Im2Vec only on the digit Zero. For GRIMOIRE, we use the models trained on the full dataset conditioned on the respective class.

**Vector Conditioning.** We also evaluate GRIMOIRE on another task previously unavailable for image-supervised vector graphic generative models, which is text-guided icon completion. Figure 6 shows the capability of our model to complete an unseen icon, based on a set of given context strokes that start at random positions. GRIMOIRE can meaningfully complete various amounts of contexts, even when the strokes of the context stem from disconnected parts of the icon. We provide a quantitative analysis in Section A.10. The results in this section are all obtained with the default pipeline that post-processes the generation of our model. A detailed analysis of our post-processing is provided in Section A.5 and Section A.6.

## 5.3 FLEXIBILITY

Finally, we demonstrate the flexibility of GRIMOIRE through additional qualitative results on new SVG attributes. One of the advantages of splitting the generative pipeline into two parts is that the ART module can be fully decoupled from the visual attributes of the SVG primitives. Instead, the vector prediction head of the VSQ can be extended to include any visual attribute supported by the differentiable rasterizer. Specifically, we activate the prediction heads $\Phi_{\text{width}}$ and $\Phi_{\text{color}}$ —outlined in Section 3.1— to enable learning of stroke width and color, respectively. We train the VSQ module on input patches while varying the values of those attributes and present the qualitative outcomes in Figure 7, where each stroke is randomly colored using an eight-color palette and a variable stroke width. The VSQ module accurately learns these features without requiring altering the size of the codebook or modifying any other network configurations.

A similar analysis is conducted with closed shapes, and the results are reported in Figure 8, showing that the VSQ module jointly maps both shape and color to a single code. This highlights the minimal requirements of GRIMOIRE in supporting additional SVG features. In contrast, other state-of-the-art vector-based generative models often rely on complex tokenization pipelines, making the extension to new SVG attributes more cumbersome and less flexible.

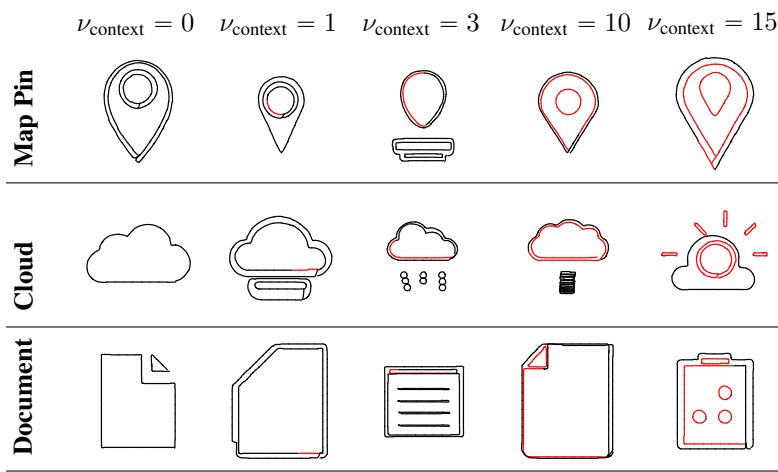

Figure 6: Different completions with varying number of context segments $\nu_{\text{context}}$ (marked in red). GRIMOIRE can meaningfully complete irregular starting positions of the context strokes.

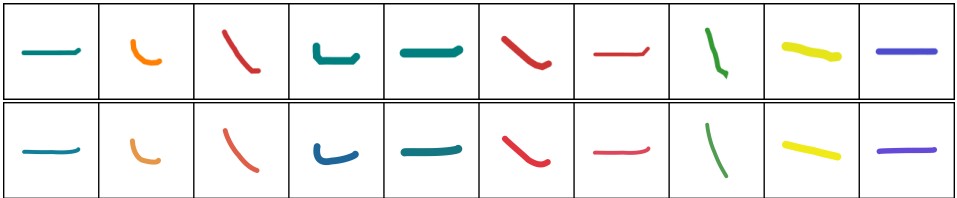

Figure 7: Inputs (top) and corresponding reconstructions (bottom) generated by a VSQ model trained to predict not only the shape but also the visual attributes of the input strokes, such as color and stroke width. Input from the test-set.

## 6 CONCLUSION

This work presents GRIMOIRE, a novel framework for generating and completing complex SVGs, trained solely on raster images. GRIMOIRE improves existing raster-supervised SVG generative networks in output quality, while offering significantly greater flexibility through text-conditioned generation. We validate GRIMOIRE on filled shapes using a simple tile-patching strategy to create the input data, and on strokes using fonts and icons datasets. Our results demonstrate the superior performance of GRIMOIRE compared to existing models, even when adapted to specific image classes. Additionally, we show that GRIMOIRE can be seamlessly extended to support new SVG attributes when included in the training data.

Future work could explore incorporating additional vector primitives, expanding visual features, or employing a hierarchical approach to patch extraction.

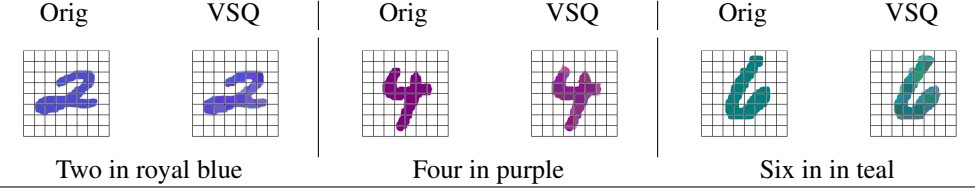

Figure 8: Reconstruction of MNIST digits when the VSQ module also predicts the filling color. The left side shows the tiling of the original raster images, the right side reports the reconstructions from the VSQ module. No post-processing is applied. Input from the test-set.

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

# A APPENDIX

## A.1 EXAMPLES OF SEGMENTATION-GUIDED PATCH EXTRACTION

In this section, we provide example results on emoji generation using some of the options mentioned in Section 6. The model setup is similar to the experiments presented for the MNIST dataset with one fundamental difference: each predicted closed shape targets one layer of the entire image canvas instead of a tile. This setting enables the prediction of a final SVG that resembles real-world use cases where vector data is a set of editable layers, ultimately composited altogether. Our training data is created using the Segment Anything (SAM) model from Meta, which provides a series of masks for the entire image. In our extraction pipeline, each mask produces one layer. We quantize the original image into 4,096 possible colors and create a raster layer for each mask by using the median color in the original image for its respective mask. A qualitative example of the results from the extraction pipeline is shown in Figure 9. The image also depicts a three-dimensional visualization of the final extracted layers sorted by their area.

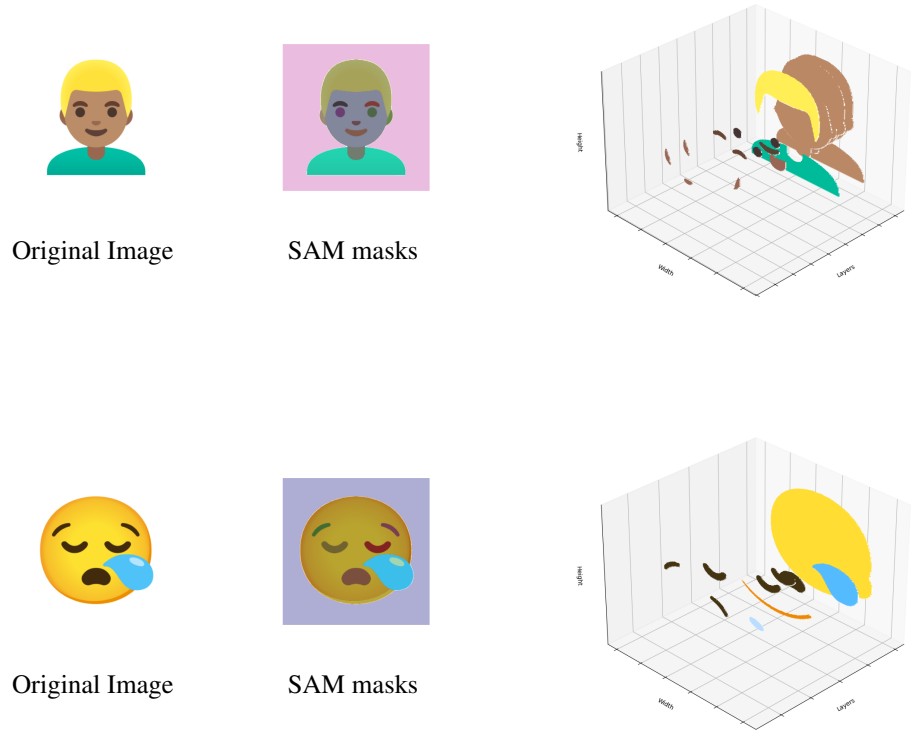

Original Image      SAM masks

Original Image      SAM masks

Figure 9: Layer extraction with SAM.

Each layer is center-cropped based on the bounding boxes of the SAM mask. A 10-pixel white padding is added on all sides similarly to what was done for the MNIST. However, in this scenario, padding does not create any artifact and merely becomes an additional scaling factor, since the reconstructed shapes fit the whole image size. During VSQ training, the ground truth cropping bounding boxes are used to scale and shift back the points predicted by the VSQ into the original position. These shifting values and the hierarchy of the layers become the new target of the ART module. We introduced minor additional changes to cope with the increasing complexity of the data, especially the color imbalance due to the small number of samples: The VSQ module outputs RGB colors per shape, but the raster and ground truth layers are converted into the CIE-LAB color

space before computing the loss. The color channels (AB) of each layer are weighted inversely to the frequency of the target color in the dataset. No weights are needed for the luminance channel. Figure 10 reports some examples of VSQ reconstructions by layer.

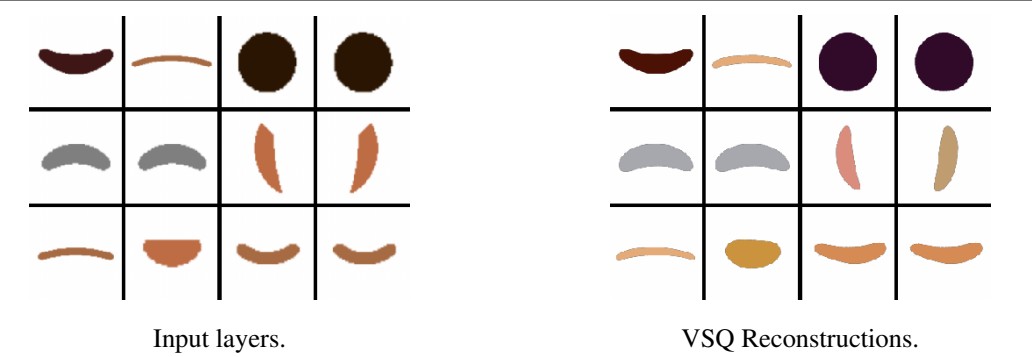

Input layers.                                    VSQ Reconstructions.

Figure 10: Inputs (left) input layers for the VSQ, (right) reconstructions of the model.

Finally, Figure 11 shows some results after compositing all the layers together. Notably, this reconstruction was achieved after training on only 110 emojis, and the results come from the test set. Common shapes (such as circles) and colors (such as yellow) are quickly learned, whereas more complicated shapes remain challenging (e.g., shapes of the hair). Overall, this is already a large improvement to other raster-supervised SVG generative models. Im2Vec does not learn the colors. As stated in the original paper and found in the repository, the colors are hard-coded to reflect the target image (e.g., one yellow and three black shapes when the target is a simple emoji).

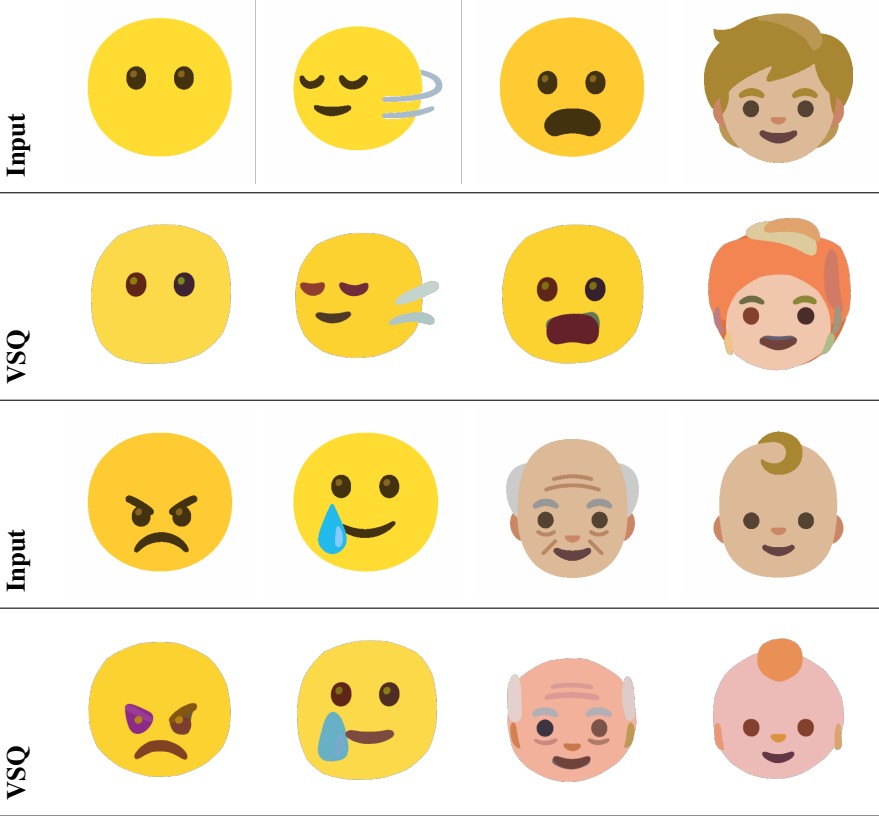

Figure 11: Reconstructions of emojis from the our VSQ, all the SVG layers are rendered together.

## A.2 KEY DIFFERENCES WITH SDS METHODS

In this section, we provide clarifications on how our model differs from popular architectures based on Score Distillation Sampling (SDS).

**Lack of target.** SDS methods do not involve training, and rely on pretrained backbones (often diffusion models), which produce more artistic and visually-appealing results, but also unbound to any specific target data. In other words, SDS methods lack any control on the target domain. To highlight this aspect, Figure 12, Figure 13, and Figure 14 reports examples of class images adopted in this work, and shows the different generations obtained with with GRIMOIRE and popular SDS methods such as VectorFusion and CLIPDraw. GRIMOIRE produces simple yet diverse generations, which are coherent with its reference dataset. In contrast, in all cases, the generations from SDS based methods appear distant from the target distribution, often partially ignoring the "black and white" suffix in the prompt, obviating the need for a more in-depth comparison with the results of our work.

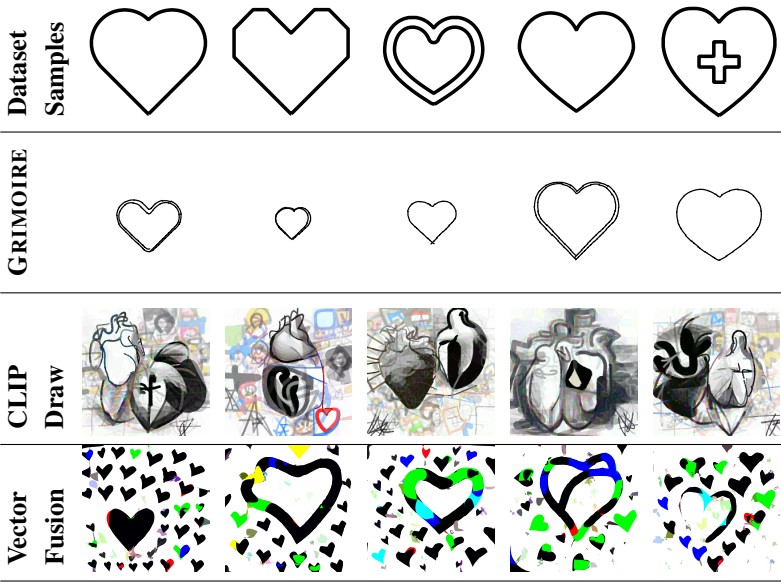

Figure 12: Comparison between SVGs generated with GRIMOIRE and methods based on Score Distillation Sampling (SDS) on the class "Heart". The first row reports samples from FIGR-8. For CLIPDraw and VectorFusion we used the prompt: "The icon of an heart, black and white".

**Speed.** SDS methods are also iterative by design; this means that generating results is extremely slow. One of the motivations behind training a generative pipeline like GRIMOIRE is that at inference time, producing a new sample merely takes the time of a forward pass. Indeed, GRIMOIRE results in two orders of magnitude faster than popular SDS based methods. In Table 4, we report the generation time (time for inference and file saving) for an image with Grimoire, VectorFusion, and CLIPdraw. Those values were obtained across 5 generations on one NVIDIA H100.

| Model | Generation Time (seconds) |
| --- | --- |
| GRIMOIRE (ART module) | **2.34** |
| CLIPDraw | 100.19 |
| VectorFusion | 379.74 |

Table 4: Average generation times of an icons given only the text prompt measured on five samples using one NVIDIA H100 GPU. SDS-based methods are extremely slow due to their iterative optimization strategy and result impractical for real-life applications.

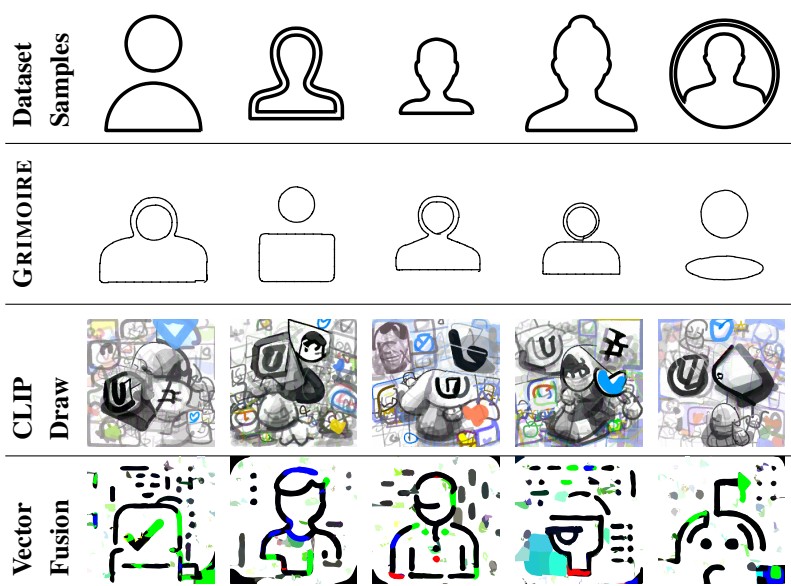

Figure 13: Comparison between SVGs generated with GRIMOIRE and methods based on Score Distillation Sampling (SDS) on the class "User". The first row reports samples from FIGR-8. For CLIPDraw and VectorFusion we used the prompt: "Icon of a user, black and white".

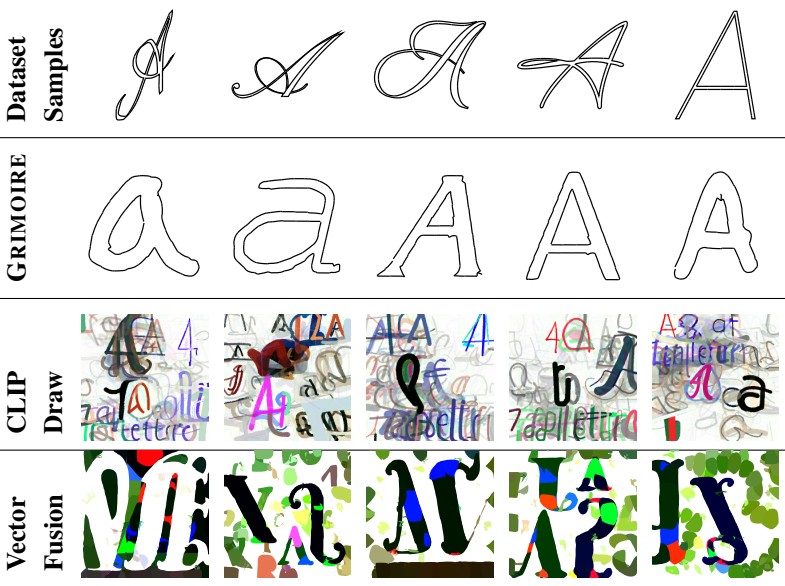

Figure 14: Comparison between SVGs generated with GRIMOIRE and methods based on Score Distillation Sampling (SDS) on the letter "A". The first row reports samples from Fonts. For CLIPDraw and VectorFusion we used the prompt: "The letter A, font".

### A.3  DIAGRAM FOR VSQ

In Figure 15, we have included a new figure depicting the VSQ module in detail.

The image shows the training pass on a stroke input patch when both the geometric and the reconstruction losses are enabled. In the figure, the architecture predicts both the Bézier curve, the stroke

width, and the colors. All of these aspects are optimized by the reconstruction loss, whereas the point coordinates are also subjected to geometric regularization.

In the shown example, we assume to use $\xi$ codes to reconstruct one single shape. In practice, for our experiments, we keep $\xi = 1$ and disable the product of all codes that create the final embedding for the decoder.

We report the size of each embedding of the autoencoder in the top-right corner and use the notation adopted earlier in the paper.

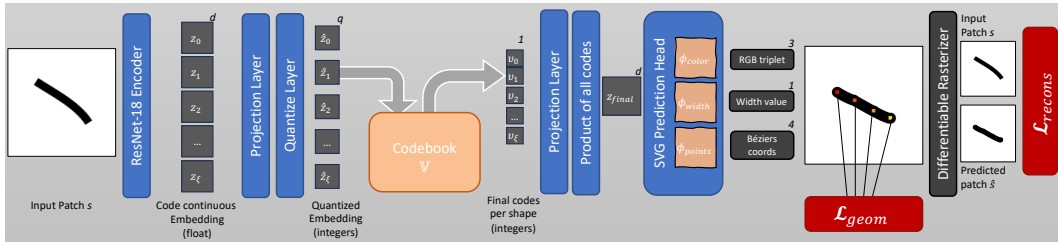

Figure 15: Overview of the VSQ pipeline with additional details.

## A.4  PRE-PROCESSING

This section provides additional information regarding the pre-processing and extraction techniques on the employed datasets.

**Shapes.** No pre-processing is conducted for the MNIST dataset. Images are simply tiled using a $6 \times 6$ grid and the central position of each tile in the original image is saved.

**Strokes.** For the FIGR-8 dataset, the pixels outlining the icons are isolated using a contour finding algorithm (Lorensen & Cline, 1987) and the coordinates are then used to convert them into vector paths. This simple procedure available in our code repository allows us to efficiently apply a standard pre-processing pipeline defined in Carlier et al. (2020) and already adopted by other studies (Wu et al., 2023; Tang et al., 2024). The process involves normalizing all strokes and breaking them into shorter units if their length exceeds a certain maximum percentage of the image size. Finally, each resulting path fragment is scaled, translated to the center of a new canvas $s$ by placing the center of its bounding box onto the center of $s$, and rasterized to become part of the training data. Since strokes in $S$ are all translated around the image center, the original center position $\theta$ of the bounding box in $I$ is recorded for each $s$ and saved. These coordinates are discretized in a range of $256 \times 256$ values. This approach is also used for Fonts, but since the data comes in vector format, there is no need for contour finding.

## A.5  POST-PROCESSING

Our approach introduces small discrepancies with the ground truth data during tokenization. The VSQ introduces small inaccuracies in the reconstruction of the stroke, and the discretization of the global center positions may slightly displace said strokes. The latter serve as the training data for the auto-regressive Transformer and therefore represent an upper limit to the final generation quality. Similarly for MNIST, the use of white padding on each patch to facilitate faster convergence results in small background gaps when rendering all shapes together, as shown in Figure 5. These small errors compound for the full final image and may become fairly visible in the reconstructions.

While we opted not to modify the global reconstructions of MNIST generation, for FIGR-8 and Fonts, we make use of SVG post-processing similar to prior work (Tang et al., 2024), which introduced Path Clipping (PC) and Path Interpolations (PI). In PC, the beginning of a stroke is set to the position of the end of the previous stroke. In PI, a new stroke is added that connects them instead. As we operate on visual supervision, the ordering of the start and end point of a stroke is not consistent. Hence, we adapt these two methods to not consider the start and end point, but rather consider the nearest neighbors of consecutive strokes. We also add a maximum distance parameter to the post-processing in order to avoid intentionally disconnected strokes to get connected. See



Figure 16: Different SVG post-processing methods visualized. From left to right: raw generation, results of applying PC and PI, results of applying PC and PI by only considering nearest neighbors of consecutive strokes.

Figure 16, Figure 17 for a qualitative depiction of this process and Section A.6 for a quantitative comparison.

|  | capital i in regular font | c in regular font | Capital l in regular font | capital r in bolditalic font | 2 in italic font | 4 in normal font |
|---|---|---|---|---|---|---|
| **Unfixed Pred.** | i | C | L | R | 2 | 4 |
| **PI Fixing** | i | C | L | R | 2 | 4 |
| **PC Fixing** | i | C | L | R | 2 | 4 |

Figure 17: Some examples of text-conditioned glyph generation from GRIMOIRE. The first row shows the unfixed model predictions, the second and third rows depict the final outputs with two different post-processing techniques.

### A.6 RESULTS WITH DIFFERENT POST-PROCESSING

In GRIMOIRE, the resulting full vector graphic generation is characterized by fragmented segments. This is because the output strokes of the VSQ decoder are each locally centered onto a separate canvas, and the auto-regressive Transformer, which is responsible for the absolute position of each shape, returns only the center coordinates of the predicted shape without controlling the state of connection between different strokes. To cope with this, in Section A.5, we introduced several post-processing algorithms. In this section, we report additional information about the performance of each of them for the VSQ module (reconstruction) and the overall GRIMOIRE (generation). Table 5 shows that the PC technique consistently outperforms the alternatives across both datasets in terms of both FID and CLIPScore.

### A.7 IM2VEC ON OTHER CLASSES

We conducted a more in-depth analysis of the generative capabilities in Im2Vec after training on single subsets of FIGR-8, and compare the results with GRIMOIRE. We trained Im2Vec on the top-10 classes of FIGR-8: Camera (8,818 samples), Home (7,837), User (7,480), Book (7,163), Clock (6,823), Flower (6,698), Star (6,681), Calendar (misspelt as *caledar* in the dataset, 6,230), and Document (6,221). Table 6 compares the FID and CLIPScore with GRIMOIRE. Note that we train our model only once on the full FIGR-8 dataset and validate the generative performance

| Model | Fonts | | | FIGR-8 | | |
|---|---|---|---|---|---|---|
| | MSE | FID | CLIP | MSE | FID | CLIP |
| VSQ | 0.0144 | 4.45 | 28.61 | 0.0045 | 1.29 | 31.17 |
| VSQ (+PC) | 0.0135 | **0.23** | **29.24** | **0.0023** | 0.10 | 31.97 |
| VSQ (+PI) | **0.0106** | 0.29 | 28.96 | 0.0028 | **0.07** | **32.0** |
| GRIMOIRE | *n/a* | 4.44 | 28.45 | *n/a* | 4.20 | 26.96 |
| GRIMOIRE (+PC) | *n/a* | **1.67** | **28.64** | *n/a* | **3.58** | **27.45** |
| GRIMOIRE (+PI) | *n/a* | 1.86 | 28.43 | *n/a* | 4.57 | 26.73 |

Table 5: Reconstruction capabilities of our VSQ module and generative performance of GRIMOIRE with different post-processing techniques after training on Fonts and FIGR-8.

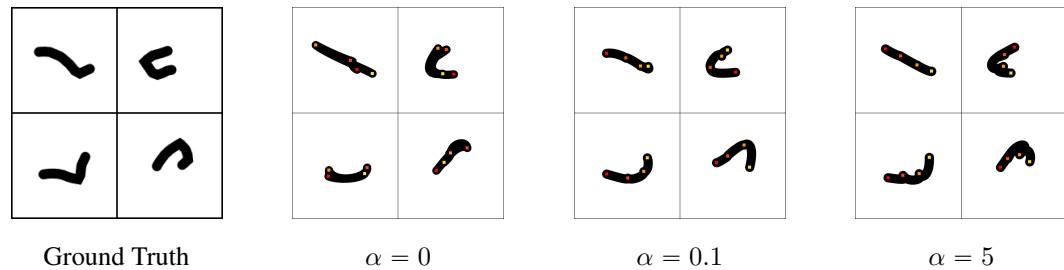

| Ground Truth | $\alpha = 0$ | $\alpha = 0.1$ | $\alpha = 5$ |
|---|---|---|---|

Figure 18: Samples from the test set when training the VSQ module with and without our geometric constraint. Each stroke consists of two cubic Bézier segments. Embedded within each stroke, the red dots mark the start and end points, while the green and blue dot pairs are the control points of each segment.

using text-conditioning on the target class, whereas Im2Vec is unable to handle training on such diverse data. Despite Im2Vec appearing to obtain higher scores on several classes such as User or Document, a qualitative inspection reveals how the majority of the generated samples come in the form of meaningless filled blobs or rectangles. The traditional metrics employed in this particular generative field, based on the pre-trained CLIP model, react very strongly to such shapes in contrast to more defined stroke images. We refer reviewers to the qualitative samples in Table 14. We further observe a low variance in the generations when Im2Vec learns the representations of certain classes, such as star icons.

| Model | camera | | home | | user | | book | | clock | | cloud | | flower | | calendar | | document | |
|---|---|---|---|---|---|---|---|---|---|---|---|---|---|---|---|---|---|---|---|
| | FID | CLIP | FID | CLIP | FID | CLIP | FID | CLIP | FID | CLIP | FID | CLIP | FID | CLIP | FID | CLIP | FID | CLIP |
| Im2Vec (filled) | 9.21 | 27.86 | **3.48** | 26.85 | **2.12** | **28.92** | 7.18 | **27.26** | 6.12 | 26.38 | 17.43 | 24.38 | **6.61** | **25.42** | **4.5** | **27.26** | 12.19 | **28.65** |
| Im2Vec | 9.05 | 27.18 | 9.19 | 25.95 | 6.33 | 27.01 | 8.63 | 25.84 | **5.09** | 25.69 | 25.58 | 24.38 | 6.8 | 23.34 | 6.61 | 26.22 | 16.62 | 26.71 |
| GRIMOIRE | 6.74 | 29.81 | 7.16 | 27.16 | 5.45 | 26.81 | 6.65 | 27.1 | 7.22 | 26.32 | 6.78 | 24.96 | 10.27 | 22.00 | 5.57 | 26.23 | 4.08 | 27.96 |
| GRIMOIRE (+PC) | **5.77** | **30.22** | 7.6 | **27.41** | 4.38 | 27.18 | **5.8** | 27.24 | 6.79 | **26.45** | **6.05** | **25.51** | 9.37 | 22.46 | 5.09 | 26.41 | **3.81** | 28.21 |
| GRIMOIRE (+PI) | 7.5 | 29.46 | 7.44 | 27.01 | 5.95 | 26.85 | 6.79 | 27.08 | 7.63 | 26.12 | 7.09 | 24.73 | 9.97 | 22.04 | 5.87 | 25.98 | 4.21 | 27.89 |

Table 6: Quality of generations for GRIMOIRE and Im2Vec for the top-10 classes in FIGR-8.

## A.8 QUALITATIVE RESULTS OF THE GEOMETRIC LOSS

The adoption of our geometric constraint improves the overall reconstruction error, which we attribute to the network being encouraged to elongate the stroke as much as possible. The results in Figure 18 show the effects on the control points of the reconstructed strokes from the VSQ. With the geometric constraint, the incentive to stretch the stroke works against the MSE objective, which results in an overall longer stroke and therefore in greater connectedness in a full reconstruction and an overall lower reconstruction error. We also present an example with an excessively high geometric constraint weight ($\alpha = 5$) demonstrating that beyond a certain threshold, the positive effect diminishes, resulting in degenerated strokes.

| Model | Fonts | | FIGR-8 | |
|---|---|---|---|---|
| | FID | CLIP | FID | CLIP |
| GRIMOIRE (w/o context) | **1.67** | **28.64** | **3.58** | **27.45** |
| GRIMOIRE (+ 3 stroke context) | 2.78 | 27.25 | 4.65 | 25.31 |
| GRIMOIRE (+ 6 stroke context) | 3.16 | 27.25 | 5.46 | 25.54 |
| GRIMOIRE (+ 12 stroke context) | 2.95 | 27.57 | 6.04 | 25.85 |
| GRIMOIRE (+ 24 stroke context) | 2.25 | 28.12 | 6.05 | 26.39 |

Table 7: Generation quality of GRIMOIRE with different lengths of provided context on Fonts and FIGR-8. Post-processing is conducted for all setups. GRIMOIRE uses textual input for all generations.

## A.9 IMPLEMENTATION DETAILS

We use AdamW optimization and train the VSQ module for 1 epoch for Fonts and FIGR-8 and five epochs for MNIST. We use a learning rate of $\lambda = 2 \times 10^{-5}$, while the auto-regressive Transformer is trained for $\sim$30 epochs with $\lambda = 6 \times 10^{-4}$. The Transformer has a context length of 512. Before proceeding to the second stage, we filter out icons represented by fewer than ten or more than 512 VSQ tokens, which affects 12.16% of samples. We use p-sampling for our generations with GRIMOIRE. Training the VSQ module on six NVIDIA H100 takes approximately 48, 15, and 12 hours for MNIST, FIGR-8, and Fonts, respectively; the ART module takes considerably fewer resources, requiring around 8 hours depending on the configuration. Regarding Im2Vec, we replace the Ranger scheduler with AdamW (Loshchilov & Hutter, 2017) and enable the weighting factor for the Kullback–Leibler (KL) divergence in the loss function to *0.1*, as it was disabled by default in the code repository, preventing any sampling. We train Im2Vec with six paths for 105 epochs with a learning rate of $\lambda = 2 \times 10^{-4}$ with early stopping if the validation loss does not decrease after seven epochs. Regarding the generative metrics, we utilized CLIP with a ViT-16 backend for FID and CLIPScore.

## A.10 GENERATIVE SCORES WITH COMPLETION

To evaluate if GRIMOIRE generalizes and learns to meaningfully complete previously unseen objects, we compare the CLIPScore and FID of completions with varying lengths of context. The context and text prompts are extracted from 1,000 samples of the test set of the FIGR-8 dataset. The results are shown in Table 7.

While GRIMOIRE can meaningfully complete unseen objects, the quality of these completions is generally lower than the generations under text-only conditioning. This is expected, as prompts in the test set are also encountered during training (the class names). The CLIPScore generally drops to its lowest point with the least amount of context and then recovers when more context is given to the model, which coincides with our qualitative observations that with only a few context strokes, GRIMOIRE occasionally ignores them completely or completes them in an illogical way, reducing the visual appearance.

## A.11 DOMAIN TRANSFER CAPABILITIES FOR RECONSTRUCTION

To validate how the strokes learned during the first training stage adapt to different domains, we use our VSQ module to reconstruct Fonts after training on FIGR-8, and vice versa. Figure 19 provides a qualitative example for each setting. Despite the loss value for each image being around one order of magnitude higher than the in-domain test-set (MSE$\approx 0.05$), the VSQ module uses reasonable codes to reconstruct the shapes and picks curves in the correct directions. Straight lines end up being the easiest to decode in both cases.

## A.12 CODEBOOK USAGE FOR STROKES

As described in Section 3.1, for FSQ, we fixed the number of dimensions of the hypercube to 5 and set the individual number of values for each dimension as $L = [7, 5, 5, 5, 5]$ for a total codebook size of $|B| = 4{,}375$. In this section, we want to share some interesting findings about the learnt

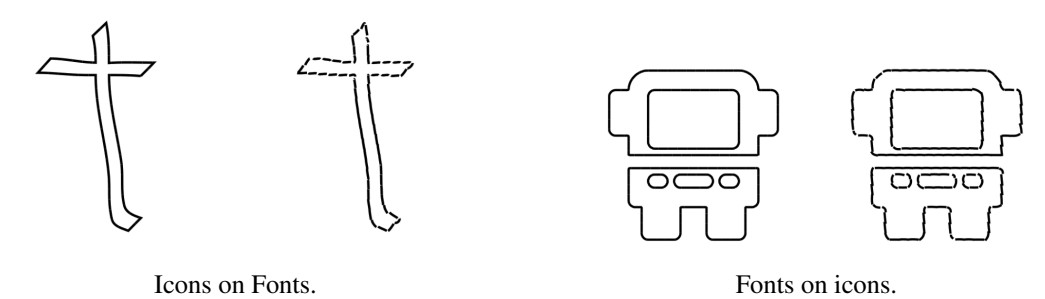

Icons on Fonts.                                    Fonts on icons.

Figure 19: Qualitative zero-shot reconstructions from the test-set of FIGR-8 and Fonts after training the VSQ module solely on the respective other dataset.

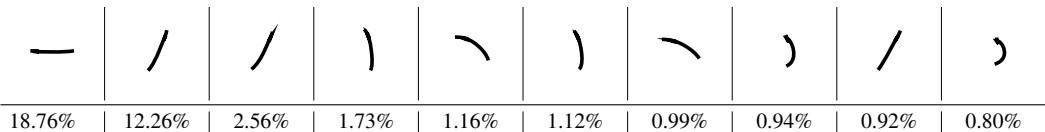

| 18.76% | 12.26% | 2.56% | 1.73% | 1.16% | 1.12% | 0.99% | 0.94% | 0.92% | 0.80% |

Table 8: Top ten most used strokes of the VSQ module trained on icons and their relative occurrences in our subset of FIGR-8.

codebook. For this, we shall use the VSQ trained on FIGR-8 with $n_{\text{code}} = 1$, $n_{\text{seg}} = 2$, a maximum stroke length of 3.0, and the geometric constraint with $\alpha = 0.2$.

After training the VSQ on FIGR-8, we tokenize the full dataset. The resulting VQ tokens stem from 60.09% of the codebook, while 39.91% of the available codes remained unused. The ten most used strokes make up 41.24% of the dataset, while the top 24 and 102 strokes make up roughly 50% and 75%, respectively. These findings indicate that for these particular VSQ settings, one could experiment with smaller codebook sizes.

To balance out the stroke distribution, one could use a different subset of FIGR-8. Currently, the classes "menu", "credit card", "laptop", and "monitor" are contributing the most to the stroke imbalance, with 26%, 24.3%, 24.05%, and 23.8% of their respective strokes being the most frequent horizontal one in Table 8.

### A.13 AVERAGE STROKES IN CODEBOOK

In Section A.12, we show the ten most used strokes of our trained VSQ, but after inspecting the full codebook we notice how neighboring codes often express very similar strokes. Therefore, to visualize the codebook more effectively, we plot mean and minimum reductions of the full codebook in Figure 20. Additionally, we tokenize the full FIGR-8 dataset and plot the same reductions in Figure 21 to show the composition of the dataset.

### A.14 QUALITATIVE RESULTS – RECONSTRUCTION

In Table 9 and Table 10, we provide several qualitative examples of vector reconstructions using Im2Vec and our VSQ module on the Fonts and FIGR-8 datasets, respectively. We fill the shapes of the images when using Im2Vec, since the model creates SVGs as series of filled circles and would not be able to learn from strokes with a small width. Im2Vec does not converge when trained on the full datasets, whereas it returns some approximate reconstruction of the input when only a single class is adopted. In contrast, the VSQ module generalizes over the full dataset.

### A.15 QUALITATIVE RESULTS – GENERATION

In this section, we provide qualitative examples of our reconstruction and generative pipeline, and compared those with Im2Vec. Table 11 reports a few examples of icons generated with GRIMOIRE

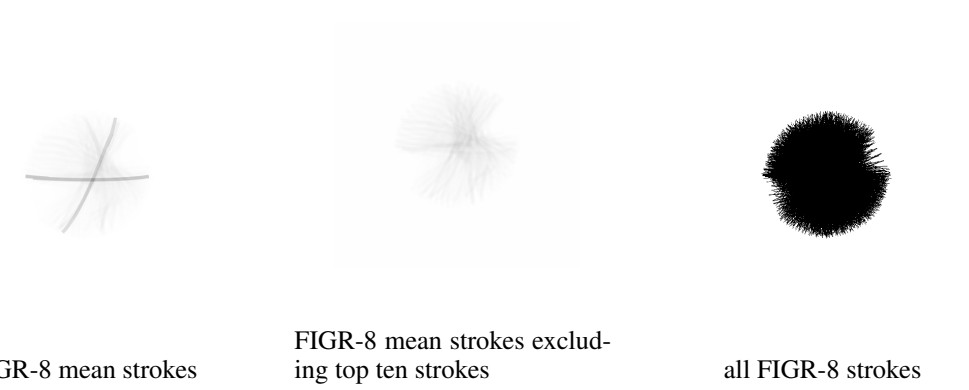

codebook mean strokes        all codebook strokes

Figure 20: Different reductions of all 4,375 strokes from the VSQ codebook. The model seems to have learned an expressive codebook-decoder mapping as the figure on the left shows a smooth and evenly distributed stroke profile and the figure on the right displays strokes in almost every direction.

FIGR-8 mean strokes      FIGR-8 mean strokes excluding top ten strokes      all FIGR-8 strokes

Figure 21: Different reductions of all strokes from the tokenized FIGR-8 dataset. The visualization on left shows the dominance of the two most occurring strokes, the middle shows that the distribution of strokes is skewed. The missing 39.91% of strokes are also visible in the right figure, where certain diagonal strokes that are available in the codebook are never used.

using only text-conditioning on classes. In Table 12 we report some generations for MNIST. In Table 13, we report generative results for Fonts. Thanks to the conditioning, we can generate upper-case and lower-case glyphs in bold, italic, light styles, and more. As can be seen in the table, GRIMOIRE also learns to properly mix those styles only based on text. Finally, in Table 14, we report some generative results on icons and Fonts for Im2Vec on a single class dataset. The results show how the pipeline typically fails to produce meaningful or sufficiently diverse samples.

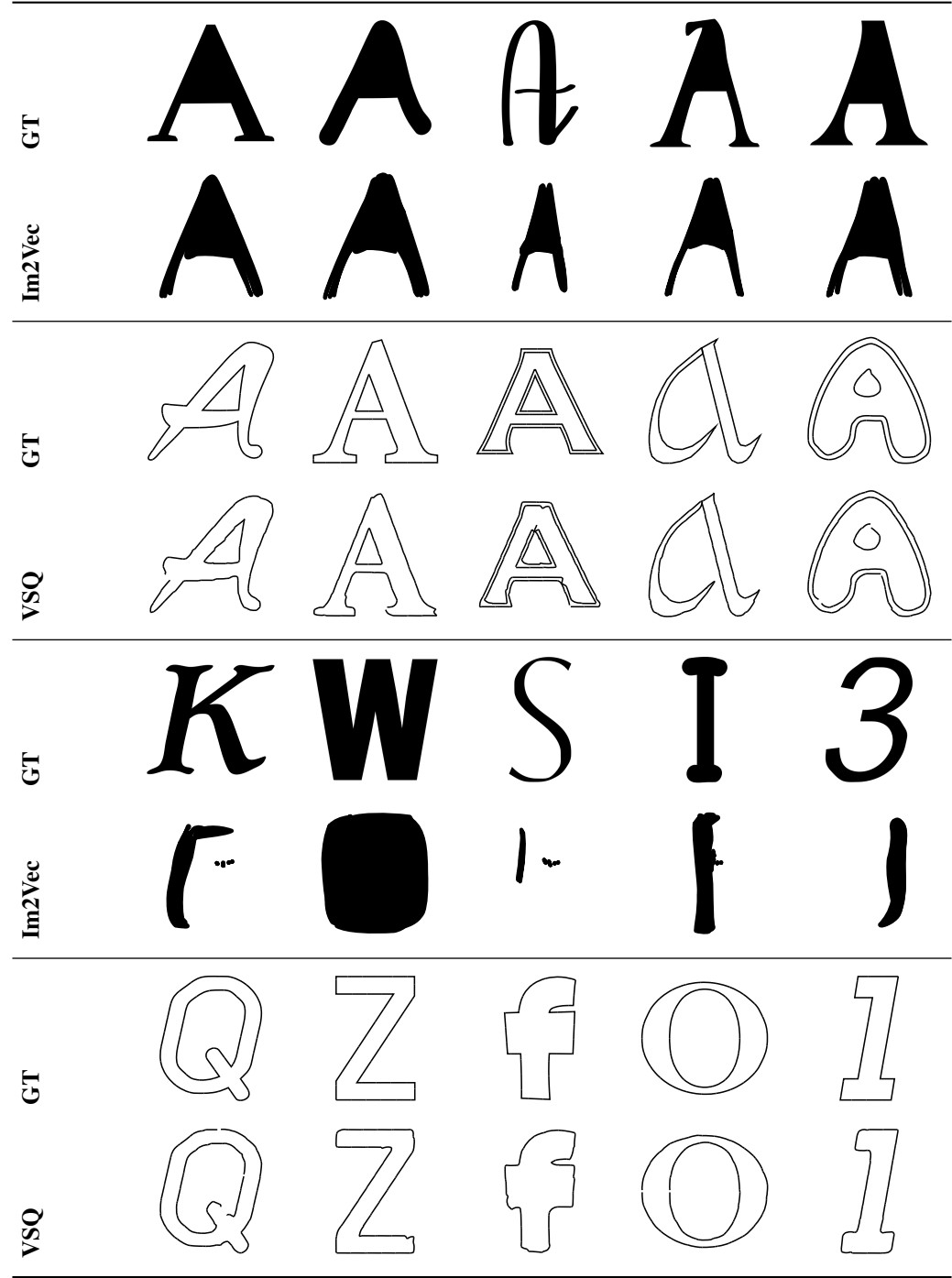

Table 9: Examples of various reconstructions of our VSQ module after training on Fonts compared to reconstructions of Im2Vec trained on the letter "A" (first row) and Im2Vec trained on the full Fonts dataset (third row).

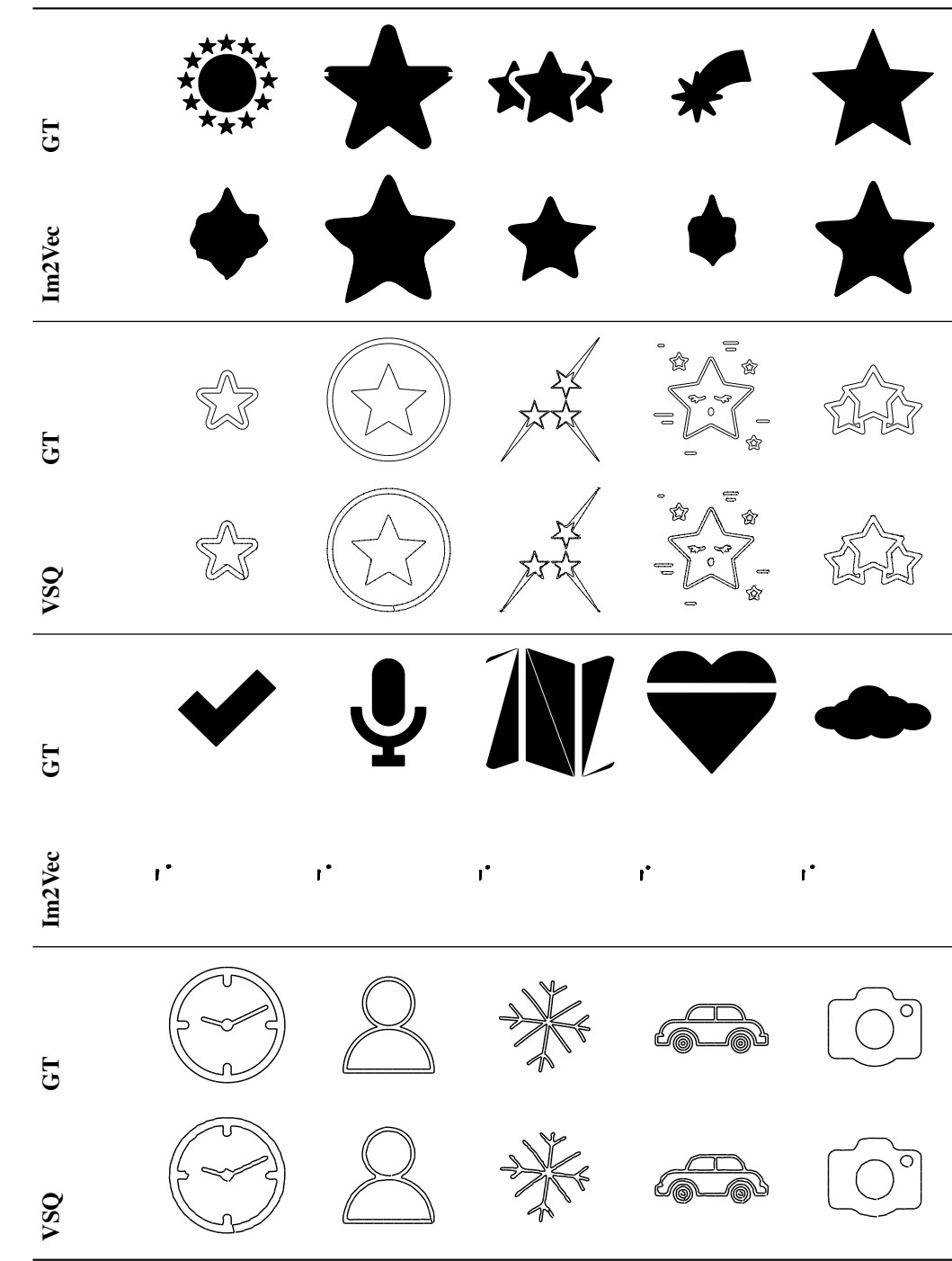

Table 10: Examples of various reconstructions of our VSQ module after training on icons compared to reconstructions of Im2Vec trained on one class (first row) and Im2Vec trained on the full dataset (third row).

## A.16 GLOSSARY OF NOTATION

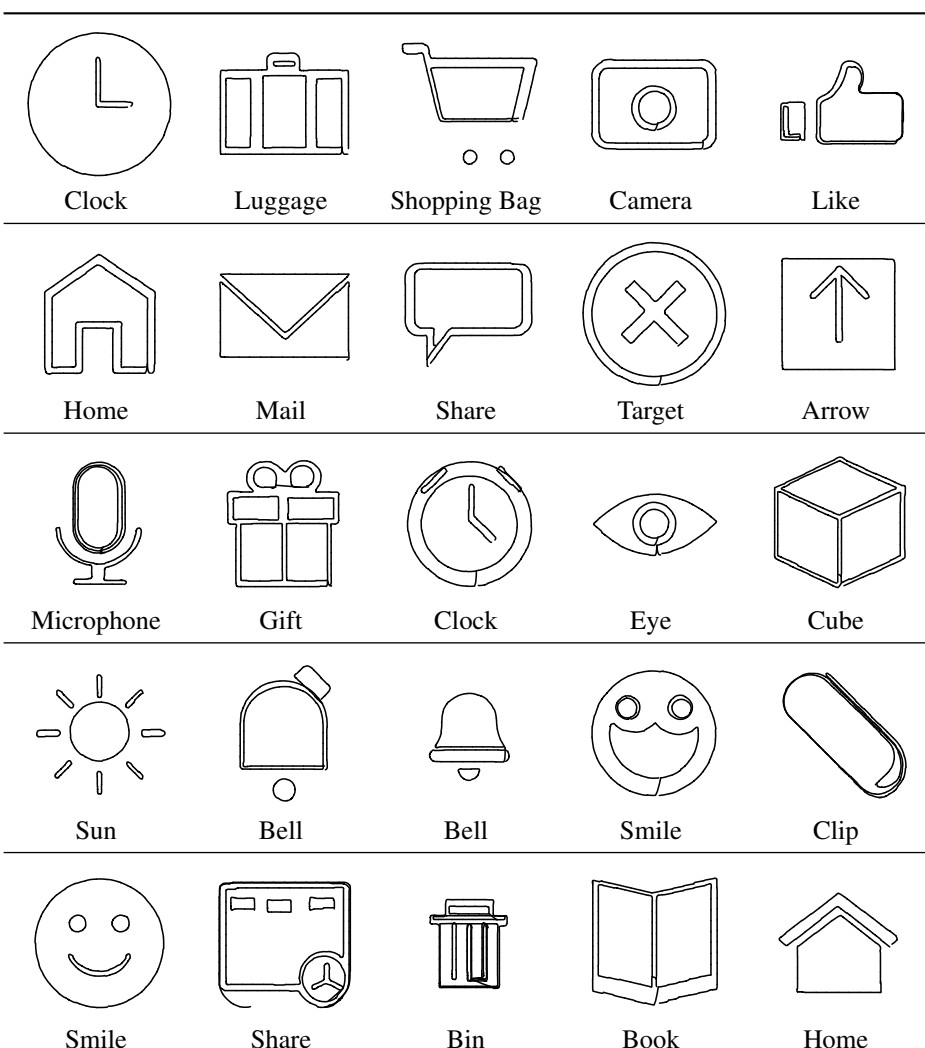

Table 11: Examples of various samples generated with GRIMOIRE after training on icons, using only text conditioning.

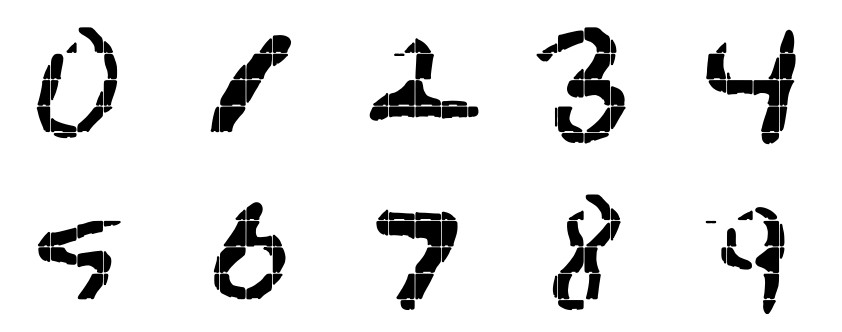

Table 12: Examples of a samples generated with GRIMOIRE for each digit of the MNIST dataset.

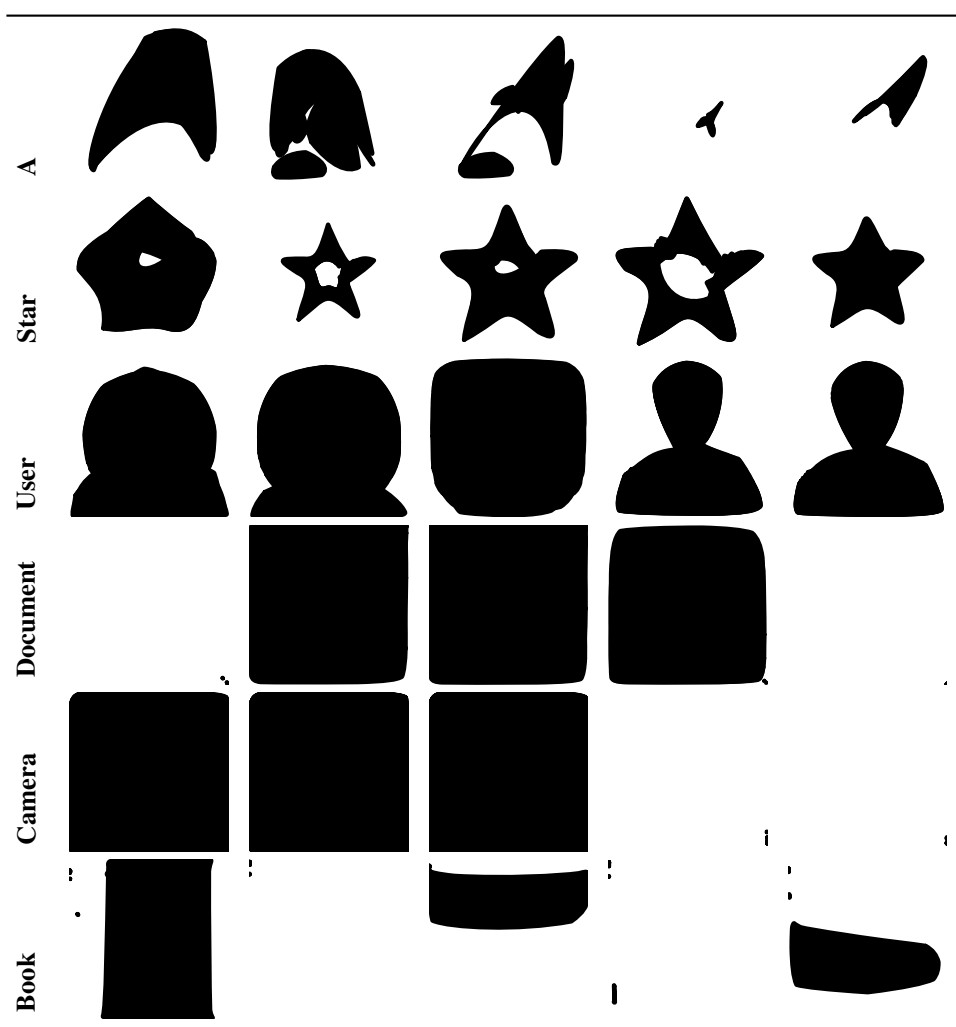

Table 13: Examples of filled samples generated with Im2Vec after training the model on specific classes of the dataset. For most classes, Im2Vec could not capture the diversity of the data and failed to meaningfully converge.

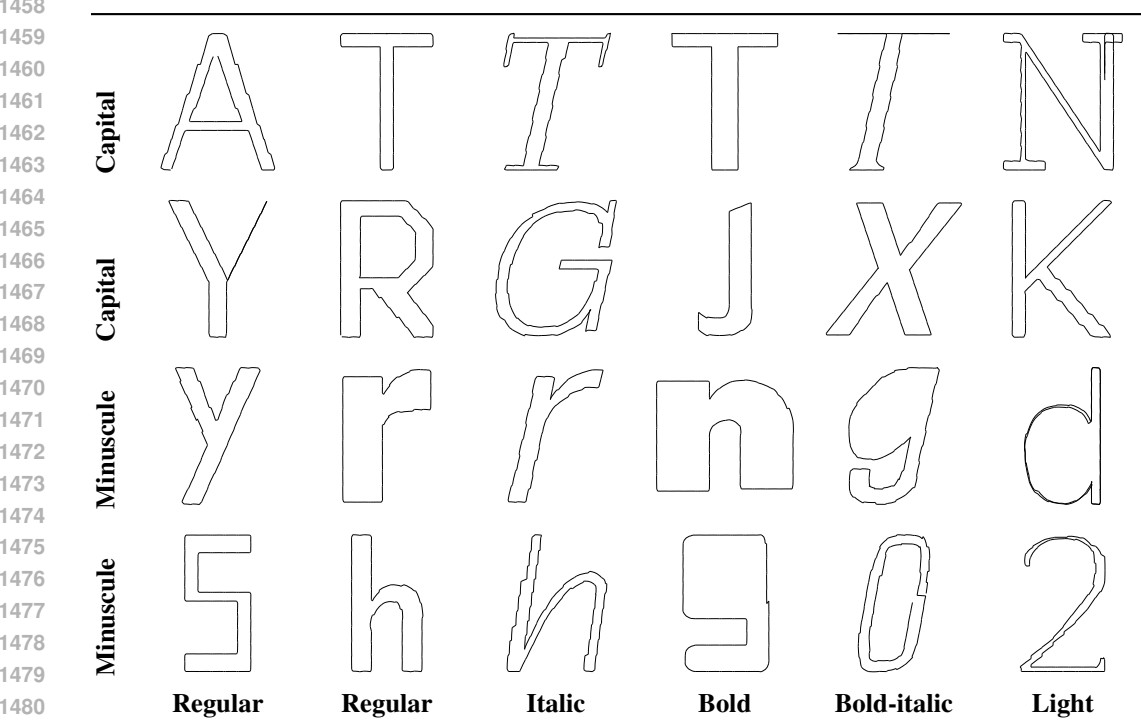

Table 14: Examples of various samples generated with GRIMOIRE after training on Fonts, using only text conditioning.

| | |
|---|---|
| $E$ | Network encoder |
| $D$ | Network decoder |
| $I$ | Image from the dataset |
| $V$ | Codebook |
| $v$ | Codes from the codebook |
| $L$ | Set of values per dimension of our codebook |
| $l$ | Single dimensional value |
| $q$ | Number of dimensions of the codebook |
| $\mathbf{S}$ | Series of patches |
| $s$ | Single patch |
| $C$ | Color channels |
| $n$ | Number of patches |
| $\Theta$ | Set of discrete coordinates |
| $\theta$ | Single coordinate pair |
| $\mathcal{Z}$ | Latent space |
| $\hat{z}$ | Projected embedding |
| $d$ | Dimension of latent |
| $z$ | Latent embedding |
| $\hat{s}$ | Predicted patch |
| $\nu$ | Number of segments |
| $P$ | Set of points |
| $p$ | Point pair |
| $\rho$ | Euclidian distance |
| $\Phi$ | Neural network |
| $\xi$ | Number of codes |
| $T$ | Text description |
| $\mathcal{T}$ | Tokenized description |
| $\tau$ | Text tokens |
| $t$ | Number of text tokens |

