# OpenReview forum: "Vector Grimoire: Codebook-based Shape Generation under Raster Image Supervision"
_ICLR.cc/2025/Conference — Submitted to ICLR 2025_

### Official Review · Reviewer_sQs2 · 2024-10-19

**Soundness:** 3
**Presentation:** 3
**Contribution:** 2
**Rating:** 6
**Confidence:** 4

**Summary:**

This paper focuses on the Scalable Vector Graphics (SVG) generation task. A text-guided SVG generative model, i.e., GRIMOIRE, is proposed to learn to draw vector graphics with only raster image supervision. The proposed model, which is formulated as the prediction of a series of individual shapes and positions, consists of two modules, one is a visual shape quantizer (VSQ), and the other is an auto-regressive transformer (ART). The effectiveness of the proposed method is demonstrated on both closed-filled shapes data and outline strokes data.

**Strengths:**

The whole pipeline considers the vector graphics generation into two stages. The first VSQ stage can be trained under raster supervision, solving the issue that existing works often require direct supervision from SVG data. The second ART stage makes it possible to apply the proposed method to support several applications, including text-to-SVG generation and SVG auto-completion. The experimental results demonstrate the effectiveness of the proposed method.

**Weaknesses:**

The main weaknesses are listed as follows.

1. The technical contribution of this paper is not very clear. Although the proposed method can be trained with only raster image supervision, the core modules, including DiffVG and VQ-VAE, come from existing works.

2. The proposed pipeline adopts different strategies for the MNIST digits and FIGR-8. I was wondering about the results using the same segmentation strategy, e.g., creating patches for both closed-filled shapes and outline strokes.

3. The current evaluation is not thorough. Either SVG reconstruction or text-to-SVG generation has been studied for several years. But this paper only compares the proposed method with one existing work, i.e., Im2vec, making the results less convincing.

4. For the ablation study, I cannot find any quantitative comparisons of the training losses for the proposed pipeline.

5. Although this paper states that some of the existing works are computationally expensive and impractical in real-world use cases, there are no detailed computational cost or running time analyses.

6. Minor issues. For the loss equation shown in L264-L266, the geometric loss should be added to the reconstruction loss, rather than itself.

**Questions:**

1. For closed paths, the current evaluations are only conducted on the MINST dataset. I was wondering about more comparisons on EMOJIS and ICONS datasets, similar to the experiments in Im2Vec.

2. How to determine the number of patches that should be used for closed-filled shapes?

3. How to determine the weight in the geometric loss?

---

> ### Author Response · Authors · 2024-11-27
>
> Thank you for the feedback on our work.
>
>
>
> **W1.** Regarding the technical novelty, please note that Grimore’s contribution is threefold: 1. A primitive-agnostic model that allows generation of many SVG attributes using raster supervision, 2. The novel use of a discrete auto-encoder architecture for vector graphic generation that simplifies the problem by deconstructing it into the learning of fundamental vector primitives that can be effectively composed to create complex graphics. 3. The introduction of the geometry loss as a new training objective function, which efficiently prevents artifacts from control points as discussed in Appendix A.8.
>
>
>
> **W2.** Thanks for the great question. We have implemented your suggestion regarding the development of a segmentation strategy to create patches, and tested this approach on more complex data.
>
>
>
> **We have added results using an additional layer-extraction method  based on segmentation masks** generated from the Segment Anything (SAM) model [1]  and **a new compositing method** that allows us to reconstruct all the layers similarly to real SVGs.  SAM provides very meaningful masks, which require just minor refinement to ensure – for instance – one single connected component per layer. Overall, the VSQ outputs look good.
>
>
>
> **We have updated the manuscript to provide some results** for these experiments, and added an extensive explanation of the segmentation-based patch extraction method (Appendix A.1). We would be happy to incorporate these results into the main paper for the camera-ready version.
>
>
>
> **W3.** We compare with Im2Vec because all other methods require vector data for training or do not support training at all (see next point). Although we agree that for some datasets it would be possible (e.g., FIGR-8 comes with an SVG version), this comparison becomes challenging on a raster-only dataset such as MNIST or emojis.
>
>
>
> **W4.** Could the reviewer clarify which loss comparison this point is referring to?
>
>
>
> **W5.** In Appendix A.2 **we now report a comparison of inference times with different SDS-based methods** showing that Grimoire is more than two orders of magnitude faster and those methods have impractical inference time (VectorFusion takes more than five minutes for one generation! Our ART module takes around two seconds by average). We report the table below for your convenience.
>
> We also show how when we want to train a neural network, **the zero-shot generations of SDS methods, despite looking more artistic, are incoherent with the dataset** and do not fit for a reasonable comparison.
>
>
>
> | model | inference time (seconds) |
> |--------------|--------------------------|
> | ART (ours) | 2.34 |
> | CLIPDraw | 100.19 |
> | VectorFusion | 379.74 |
>
>
>
>
> **W6.** Thanks for pointing this out. We have updated the revised version of the paper accordingly.
>
>
>
> In response to your questions:
>
>
>
> **Q1.** We have further **trained our VSQ model on more complex data** (emojis) based on patches extracted using the segmentation dataset. Despite training only on 120 emojis, the results on validation data look promising even on emojis with many details. Despite these results being already of **higher quality than what is reported by Im2Vec**, we also wish to mention that the **authors of Im2Vec had conveniently hardcoded the colors** in their experiments on emojis (e.g., one yellow path, two black paths for the eyes, etc.). Without this hack, their model would not converge at all.
>
>
>
> **Q2.** In general, this number is known during the training of the VSQ, which has to reconstruct each patch, and is learned dynamically by the ART module using the <EOS> code token.
>
> Specifically for the MNIST dataset, this number is fixed to the number of tiles during the VSQ training.
>
>
>
> **Q3** Since a better position of the control point translates into minor artifacts, we perform a sweep for multiple values of the weighting factor on the validation set of FIGR-8 and choose the one that produced the best reconstruction. We have reported qualitative examples of reconstruction with different weights in Appendix A.8.
>
>
>
>
> [1] Kirillov, A., Mintun, E., Ravi, N., Mao, H., Rolland, C., Gustafson, L., ... & Girshick, R. (2023). Segment anything. In Proceedings of the IEEE/CVF International Conference on Computer Vision (pp. 4015-4026).
>
> [1] We used the ViT-H default backbone.

---

> > ### Comment · Reviewer_sQs2 · 2024-11-28
> >
> > I appreciate the authors' clarifications. Most of my concerns have been addressed by the responses. I would lean to accept the paper by involving the additional evaluations and discussions in the revised version.

---

> > > ### Author Response · Authors · 2024-11-29
> > >
> > > Thank you for the positive comments, and constructive feedback and suggestions which lead to much richer analysis.
> > >
> > >
> > > We wish to confirm that we will indeed incorporate those analysis and evaluations in the main manuscript for the camera ready version.

---

### Official Review · Reviewer_Zk2S · 2024-10-26

**Soundness:** 3
**Presentation:** 3
**Contribution:** 2
**Rating:** 6
**Confidence:** 4

**Summary:**

This paper presents a method called GRIMOIRE for generating scalable vector graphics. The authors first train a visual shape auto-encoder that maps 2D images into a discrete codebook. Specifically, each images is divided into patches and then encoded and quantized into a set of discrete tokens, and then the decoder transforms the latent codes back into the latent space, followed by a neural network to predict the control points of the Bezier curves. By using the differentiable rasterization, the Bezier curves can be rendered back into 2D images for end-to-end training. Then, the second stage involves learning an auto-regressive transformer to model the joint distribution over text, positions of stroke tokens. The authors did the experiments on three different datasets, including MNIST, FONTS, and FIGR-8. Results demonstrated that the proposed auto-encoder can reconstruct the images well and the generator can generate good results conditioned on text prompts.

**Strengths:**

- The authors propose an image-supervised method to generate vector graphics, by fitting this task into a common used generation framework: vector-quantization auto-encoder with auto-regressive distribution learning. To assist auto-decoder learning, the authors also included novel components such as geometry losses.
- The authors have conducted detailed experiments to validate their method, both in terms of reconstruction and generation quality. The results also clearly showed that the proposed method can excel in the selected datasets compared with other baselines.
- The authors released the code and detailed instructions for great reproducibility.

**Weaknesses:**

- This paper focus on a relatively smaller problem: generating vector graphics on small datasets. The experiments are mostly done on simple datasets and I haven't seen more complex results. Compared with SDS-based methods, the proposed method are limited by the use of datasets and cannot produce more diverse results.
- Although the authors successfully applied the VQ-framework for generation, I think in terms of contribution, it's more like utilizing existing experience for some new tasks, so the technical contribution is not that significant for a higher score.

**Questions:**

- The description of auto-encoder part (Line 219 to Line 232) is a bit unclear for me, with several math equations omitted according to my understanding. First, the latent variable $z_i$ is of dim $d \times \xi$, then "each of the $\xi$ codes is projected to q dimensions", does this mean the results are $d \times q$ and there are $\xi$ codes for each patch? What is the shape of the codes $v_i$? How does the "transformation ensures that each unique combination of quantized values is mapped to a unique code"?
- How to make sure that strokes from different patches are connected to each other?

---

> ### Author Response · Authors · 2024-11-27
>
> Thanks for your valuable input and questions, especially about more complex results and a comparison with SDS-based methods.
>
>
>
> **W1.** Thanks for your suggestion. **We extended Grimoire to predict closed shapes as layers**, using a segmentation neural network to automatically extract the regions to model as layers. This allowed us to report **results on more complex imaging such as emojis** as suggested. The final image is obtained by compositing all layers together **similarly to standard SVGs.** We have added an extensive explanation of the approach in Appendix A.1.
>
>
>
> Moreover, we **have included a qualitative comparison of generations with SDS-based  methods** in Appendix A.2 on the revised paper. Our experiments show how even if those models generate more artistic results, they offer no control on the target data. **Compared to Grimoire, they fall short with respect to a target dataset** and make finetuning extremely hard.
> Also, **unlike Grimoire, those methods are extremely slow and impractical, requiring  up to five minutes** for a single generation. We have included this comparison in Table 4 of Section A.2 in our revised version of the paper. We report these values below for your convenience.
>
>
>
> | model | inference time (seconds) |
> |--------------|--------------------------|
> | ART (ours) | 2.34 |
> | CLIPDraw | 100.19 |
> | VectorFusion | 379.74 |
>
>
>
>
>
> **W2.** Regarding the technical novelty, please note that Grimore’s contribution is threefold: 1. A primitive-agnostic model that allows the generation of many SVG attributes using raster supervision, 2. The novel use of a discrete auto-encoder architecture for vector graphic generation that simplifies the problem by deconstructing it into the learning of fundamental vector primitives that can be effectively composed to create complex graphics. 3. The introduction of the geometry loss as a new training objective function, which efficiently prevents artifacts from control points as discussed in Appendix A.8.
>
>
>
> Regarding the questions on the method:
>
>
>
> **Q1.**
>
>
>
> > "each of the xi codes is projected to q dimensions", does this mean the results are d x q and there are xi codes for each patch?
>
>
>
> No, $d$ is the dimension being projected. **Each** of the $\xi$ codes is projected from $d$ to $q$ ($q$ being the number of dimensions of our codebook), resulting in $\xi \times q$ values per shape in total.
> We updated this part in the paper to make it more explicit that the dimension d is the one being projected.
>
> $\hat{z}_i$ is the quantized code value for one of the $i$-th of $\xi$ codes.
>
> We chose this notation instead of reporting the dimensions of all codes together because it comes handy for Equation (1). We have added a new image in Appendix A.3 that clarifies each dimension of the embeddings.
>
>
>
> > How does the "transformation ensures that each unique combination of quantized values is mapped to a unique code"?
>
>
>
> Each code $v_i$ is a single-dimensional integer value obtained according to Equation (1).
>
> Equation (1) ensures a “unique combination of quantized values” because the quantized value $\hat{z}_i$ for a given dimension $j$ of the hypercube is first multiplied by the basis value $b_ij$, which accounts for all the values of all the dimensions below in the hypercube. This ensures that the contribution of the code of each dimension is a unique value, before all contributions are summed together. We refer to the original FSQ paper for more information, but we are also available for further clarifications.
>
>
>
>
> **Q2.**
>
>
>
> > “How to make sure that strokes from different patches are connected to each other?”,
>
>
>
> Our post-processing algorithms reported in Appendix A.2 play a relevant role in making all strokes connected. However, we found the grid size that quantizes the stroke coordinates (described in Line 286) to have major importance in the smoothness of the connected strokes. Small grid sizes yield a disconnected appearance regardless of the post-processing, while higher grid sizes mitigate the need for post-processing because the points are already closer.

---

> > ### Author Response · Authors · 2024-12-01
> >
> > Dear Reviewer `Zk2S`,
> >
> > Since the End of the Rebuttal is coming very soon - only a few days left, we would like to inquire if our response addresses your primary concerns. If it does, we kindly request that you reconsider the score.
> >
> > If you have any additional suggestions, we are more than willing to engage in further discussions and make necessary improvements to the paper.
> >
> > Thanks again for dedicating your time to enhancing our paper!
> > Looking forward to your feedback.

---

### Official Review · Reviewer_ymmo · 2024-11-03

**Soundness:** 3
**Presentation:** 2
**Contribution:** 3
**Rating:** 6
**Confidence:** 4

**Summary:**

The paper addresses the problem of vector graphics image generation and proposes a novel approach to generate vector graphics images conditioned on text by using raster image supervision. The vector graphics image format is SVG, which is built using cubic bezier curves.

Existing gaps that motivate this paper are two fold: (a) Lack of abundant SVG data for training generative models for vector graphics generation, and (b) the difficulty of incorporating visual attributes such as color and strokes.

To address these challenges, the paper introduces a method to train a generative vector graphics model using just the raster images, eliminating the need for ground truth SVG training data. The proposed methods is a two-stage approach. The first stage decomposes an input raster image into primitive patches and learns a discrete codebook that maps each patch to a code. In the second stage, an auto-
regressive text-conditioned transformer is used to learn joint distribution of textual description and codes. The loss functions used are: (1) Reconstruction loss to reconstruct primitive strokes and shapes, (2)  Geometry loss for better placement of Bezier curve points,
(3) Causal loss for auto-regressive transformer resulting from causal multi-head attention in the decoder. As probably understood, this is a strongly supervised method. To summarize, the input to the method during training is an RGB image and a textual description of that image, while during inference, the method only takes a textual of an image we want the learned model to generate. At the output, we have a generated vector graphics image in a SVG format. The representation of SVG is Cubic Bezier curves, with control points and a starting point. The underlying modeling tools are vector-quantized auto-encoder for image encoding with ResNet as image encoder, DiffVG as differentiable rasterizer, and BERT to encode text, and an auto-regressive Transformer to generate strokes and curves in an auto-regressive fashion. The datasets used for experiments are: MNIST, Fonts, FIGR-8. In terms of evaluations, the paper looks at the following metrics: MSE, FID score and CLIP score.For comparisons, the method compares against one prior work from zcvpr 2021, Im2Vec.

**Strengths:**

1) The idea of using raster image supervision is novel

2) Effectiveness of the proposed method is shown with extensive experiments and ablation
studies

3) Usefulness is shown on multiple use cases (a) text-conditioned generation, (b) text-
conditioned icon completion (This task was not possible with previous methods. The
proposed framework made it possible), and (c) ability to reconstruct vector images with
visual attributes such as color and stroke width.

Overall, the paper proposes a novel idea to use raster image supervision to train a generative
model for vector graphics images. It uses vector quantized autoencoder to learn a codebook for
primitive strokes+shapes, and then trains an auto-regressive transformer model to learn a joint distribution on
textual description and input images, as represented by codebook trained in the earlier stage. The
paper also puts a good effort in making extensive experiments, although comparisons could be made stronger (see next section for comments),  to demonstrate the usefulness of the proposed method.

**Weaknesses:**

1) The paper could have been written in a better way to make it easier to understand.
Please see the following section for improvement suggestions

**Writing improvement suggestions:**

- The equation on line 265 (right side) does not make sense. I guess there is a spelling
mistake. Please number the equations so that it is easier to refer to.

- It would be helpful to have one more diagram explaining the stage-1 in more detail.
Figure-2 shows the steps at a very abstract level and it does not include all the steps e.g.
input and output of the model is not shown, the Geometry Loss mentioned in line 265 is
missing in the figure.

- The “Related Work” section in its current form is too elaborate and can be cut down by
writing it succinctly

- The “Method” section includes proposed method as well as the experimental details
together making it harder to read and understand. The experimental details such use of
contour finding algorithm (on line 215), variants used for $\Phi_{points}$ (on line 234-
236), the number of trainable parameters (on line 269), etc., can be factored out and
written in a separate section named “Experimental Setting”


2) Only one baseline (Im2Vec) was used for experiments. How about comparing to DeepSVG (from NeurIPS 2020) and ICON SHOP (from SIGGRAPH Asia 2023)? This would have made the evaluations thorough and helped get a better understanding of where the proposed approach stands, and what other gaps remain.

**Questions:**

* Are the results shown in Tables 1,2,3 and Figure 4,5,6,7,8, on the train set or the test set?

---

> ### Author Response · Authors · 2024-11-27
>
> Thank you for your valuable feedback.
>
>
>
> **W1.**
>
> -   We fixed the typo in the notation at Line 265
>
> -   We have created an *Experimental Setup* paragraph as you recommended inside the dataset section and streamlined the method section.
>
> -   We have included a new diagram in A.3 to clarify the training of the VSQ, as you suggested. We plan to further polish the figure for the camera ready.
>
> -   We can shrink the related work for the camera ready and instead incorporate into the main paper the qualitative results on emojis currently in Appendix A.1 or the new diagram.
>
>
>
>
> **W2**. Vector-supervised methods cannot be tested on the raster datasets. FIGR-8 offers an SVG version, which the authors from IconShop have used to train their model. This model has slightly better performance than Grimoire with a CLIPScore of 31.18 (vs 29.00) and FID score of 0.40 (vs 0.64).
>
> However, this comparison should be interpreted with caution, as the nature of the training signal differs significantly. Unlike raster strokes, IconShop requires vector supervision and learns directly from precise position tokens.
>
> We are working to provide you with results on DeepSVG, depending on the complexity of their tokenization pipeline we will share some updates by the end of the extended discussion period.
>
>
>
> Concerning your question:
>
>
>
> **Q1**. All the tables (Table 1, 2) and figures (Figure 7, 8) about reconstruction **use the test sets**. All Tables for generations (Table 3) use the test set for the FID score. Generated images (Figure 4, 5, 6) are not linked to any split. We have now included this information – when relevant – in the captions of the revised manuscript.

---

> > ### Comment · Reviewer_ymmo · 2024-11-28
> >
> > Thank you.
> >
> > 1) The writing improvements, I am sure, will be welcomed by other reviewer.
> > 2) Noted the input difference for IconShop, and yes, please share the comparison results for DeepSVG paper
> >
> > Noted the info on the use of test sets for Tables 1, 2 and Figures 7, 8. And yes, where relevant, please include this information in the caption of the revised manuscript.
> >
> > There will be no change in the score from my end.

---

> ### Author Response · Authors · 2024-12-02
>
> Dear Reviewer `ymmo`,
>
>
>
> We have extended our analysis by training DeepSVG on the same FIGR-8 split used for Grimoire.
>
>
>
> Unlike Grimoire, DeepSVG supports conditioning only on class identifiers rather than natural language prompts. Therefore, we assigned a unique identifier to each class in FIGR-8.
>
>
>
> The evaluation of the generative capabilities highlights that DeepSVG struggles to handle the complexity and diversity of FIGR-8, yielding **the lowest CLIPScore and FID performance among all models.**
>
>
>
> Below, we present a table with all the scores, highlighting the key features of each approach. We are willing to incorporate those additional results into our manuscript.
>
> | Model    | CLIPScore | FID   | conditioning | supervision | colors support | filling support | stroke-width support |
> |----------|-----------|-------|--------------|-------------|----------------|-----------------|----------------------|
> | DeepSVG  | 22.10     | 58.03 | class        | vector      |                | x               |                      |
> | IconShop | 31.18     | 0.40  | prompt       | vector      |                |                 |                      |
> | Im2Vec   | n.a.        | n.a.    | n.a.         | raster      |                |                 |                      |
> | Grimoire | 29.00     | 0.64  | prompt       | raster      | x              | x               | x
>
>
>
>
> We hope that our enriched analysis helps to better position our method and **ultimately addresses all your concerns about our submission.**

---

> ### Comment · Reviewer_ymmo · 2024-12-02
>
> Great! This analysis, including comparison along different axes, helps better position the proposed approach and adds a significant value to the evaluations. With that, all my concerns have been addressed. As before, I am leaning positive.

---

### Official Review · Reviewer_6k4C · 2024-11-07

**Soundness:** 3
**Presentation:** 2
**Contribution:** 2
**Rating:** 3
**Confidence:** 4

**Summary:**

This paper proposed a patch-based approach for learning a discrete codebook for a set of mini strokes/shapes, and then using an autoregressive transformer that operates on this representation to perform "SVG" reconstruction and generation.

**Strengths:**

- The overall pipeline design looks okay to me. It make sense to use a codebook to represent a set of instruction token, and an autoregressive transformer is a good way to interact with such a set of tokens. The design choice for encoding the images and the text prompts also are reasonable.
- Results look reasonable on the set of data shown, with considerable discussions.
- Code is available, and appears to be quite reproducible.

**Weaknesses:**

I am slightly out of touch with this class of problems recently, so I some of the stuff I mention below might not be that accurate or up to date. Would be happy to update that following clarifications. With that being said, I didn't get much out of this paper - both the problem studied and the technique used are far more well studied than what is depicted in this paper. I am aware of many approaches that are similar in terms of core ideas (autoregressive generation of 2D/3D content represented as a set of discrete instructions/parameters), many datasets that are far more complex than what is used in this paper, as well as multiple works that seem to provide better solutions to this problem (and not compared):

- Despite the claims in 2.1 about this work being different from other SVG generative models (and other CAD/sketch/3D content generation pipelines that follow similar idea of representing content with a set of tokenized instructions, none of which cited), this work is not doing anything that is that much different from these works. Use of vector quantization (e.g. SkexGen: CAD Construction Sequence Generation with Disentangled Codebooks) or grid based (quantized) features (e.g. ShapeFormer: Transformer-based Shape Completion via Sparse Representation) are both standard practices (just naming the first paper that came on top of my mind). In some sense, the paper is even a regression from some of these works, where the so called "SVG" representation used in this paper is not really SVG, but a patch wise curve/closed shape representation that lacks the structure of actual vector drawings.
- Despite the claim about lack of vector data, datasets like SketchGraph and DeepCAD offer more than enough data in vector/discrete formats, and datasets like Quick, Draw! (as cited briefly in this paper as well) already has a vector representation. Moreover, there are so many off the self approachs for vectorizing drawings (e.g. Deep Sketch Vectorization via Implicit Surface Extraction). With the vast abundance of such data, it just doesn't make sense to use the pseudo "SVG" representation in the paper, which has few of the benefits of actual vector drawings.
- Following the previous two points, the baselines used in the paper is both insufficient and inappropriate: of course the more wholistic representation used by Im2Vec  (which was also pretty ancient by the standard of current vision/graphics) won't have as much representation power as collection of patch-based strokes. However, it at least has a single, wholistic representation of the entire SVG, and thus is not only more flexible, but also allows various applications e.g. interpolation, editing, that is hard to achieve with the proposed method. Therefore, just showing that Im2Vec fails on some cases don't really make me buy into the usefulness of the proposed method: if the goal is to show "generative capabilities", then compare against many more recent SVG/sketch/CAD generation works; if the goal is to show that this works enables better raster-to-vector conversion and thus removes the need of curated dataset (of which there are many), then there's also many much more recent works to compare against; if the goal is to show that this patch-wise representation is a good representation, then more analysis should be done on how this is a good representation (and I don't think so, both from an intuitive level and by examining the various visual artifacts present in the qualitative examples) that is useful in practice (i.e. having at least some unique capacities of vector graphics, as opposed to just an image), then more analysis is needed (editing, interpolation, user studies, etc.).

**Questions:**

As mentioned in the weakness section, this paper really needs to be positioned more properly against (more recent) works on raster-to-vector, token-based 2D/3D generation, and the abundance of vector graphics/CAD datasets. I currently do not buy into the utility of this patch-based mini stroke/shape representation, that is far from a true SVG representation. Clarifications over these points will help sway me - after all, there is nothing that is fundamentally incorrect about the approach. I might also be wrong with some of my claims - am a bit out of touch with this topic, as mentioned, so please do point out anything that I missed.

---

> ### Author Response · Authors · 2024-11-27
>
> Thank you for your valuable and thoughtful feedback.
>
>
>
> **W1.** We would like to reemphasize the positioning of our method:
>
> Grimoire is a **scalable, effective and primitive-agnostic  generative framework** for the SVG standard that uses **only  raster supervision** for training.
>
>
>
> All the works mentioned in your review use vector supervision directly and require vector data, and CAD is even a different standard. **Grimoire learns from raster images, which is a significant difference.**
>
>
>
> Our model offers different opportunities compared to vector-supervised methods. For instance, how would you condition IconShop to your own input shape designed with a specific color or thickness? **This must be created in a vector format (aligned with the tokenizer's expected input) and constrained to the features enabled by the original vector training data.** In contrast, raster drawings offer greater flexibility for this task.
>
>
>
> However, we acknowledge your concerns about the complexity of the raster data used in this work. To address this, **we have included results on an additional patch-extraction pipeline, based on the output of the Segment Anything Model (SAM)**[1]. In this variant, each mask generated by SAM is used to extract a single raster layer. We applied this pipeline to emojis in PNG format, training Grimoire to reconstruct each shape as an independent vector layer. Finally, we composite all layers for visualization, mimicking the structure of real SVGs. **How could this data be handled with a vector-supervised model like IconShop?**
>
>
>
> **W2.** Vectorizing raster data for training a vector-supervised model is counter-intuitive and can be extremely inefficient, **especially in contexts where  new training data is constantly being produced and makes vectorization a bottleneck.**  In general, vectorizers are slow (The paper you mentioned takes at least 10 seconds per image, LIVE takes minutes per image), and their processing time must be put on top of the training time of the vector-supervised model.
>
>
>
> **W3.** After working on this project for more than a year, we can confidently claim that although the *wholistic representation* might at first glance look better, **this is actually extremely sub-optimal when using raster supervision;** at least in the early stages of the training.
>
> As we already mentioned to other reviewers, the experiments on emojis from Im2Vec had the ground truth colors hard-coded to facilitate convergence. Without this hack, their model does not work. **We overcame Im2Vec limitations by breaking down the complexity of SVGs into single, simpler components to composite together.** The two stage architecture also plays a crucial role to achieve valid generations: we let the model first focus on the visual primitives, and learn to place those onto the canvas in a second stage.
>
>
>
> We have included the additional experiments on more complex images in the Appendix A.1.
>
>
>
> [1] Kirillov, A., Mintun, E., Ravi, N., Mao, H., Rolland, C., Gustafson, L., ... & Girshick, R. (2023). Segment anything. In Proceedings of the IEEE/CVF International Conference on Computer Vision (pp. 4015-4026).
>
> [1] We used the ViT-h default backbone.

---

> > ### Comment · Reviewer_6k4C · 2024-11-28
> >
> > Thanks for the clarifications. My point was never about whether this method works under the specific input/output setting and training protocol, but whether it has any practical use, or if an improved version down the route pointed by this method is ever going to be useful.
> >
> > I’m not saying that one should use vectorizer to generate vector training data. If the goal is to reconstruct vector data, there is no need to go through all the additional training - just apply a vectorizer, they work better.
> >
> > If the goal is other tasks eg generation, then I’m yet to be convinced that the generation results of this method has any desirable properties that make it favorable over just a raster output - e.g many of the font reconstruction results are so fragmented and distorted to be usable.
> > I think my point about “wholistic representation” is completely misunderstood: I acknowledge that they are less effective, but at least they generate something usable while they do work, whereas the proposed method, while working more often, doesn’t seem to generate anything that’s remotely usable in downstream practical applications, and I still don’t have much clue how this patch wise representation has any nice properties of actual professional/artist created svgs.
> >
> > Or, the paper is still acceptable even if the problem studied has no practical uses, but the ideas are new and inspiring. But as Ive said, all techniques in this paper are somewhat traceable to existing works in related problems.

---

> ### Author Response · Authors · 2024-11-29
>
> Thank you for the further clarification on your concerns.
>
> Indeed, the goal of Grimoire is **not** to reconstruct vector data, but rather learning to **generate new images** from raster data.
>
>
>
> Regarding the practical uses of the generation capabilities, we would like to draw your attention to the results in Appendix A.1, which involve more complex data and realistic SVG representations. To complement this, **we wish to share with you some samples of these new reconstructions**. We have uploaded those SVGs into the “ Showcase” folder in our repository.
>
> Here is a link for your convenience: [LINK](https://github.com/under-review-papercode/9973/tree/main/Showcase/Emoji)
>
>
>
> We hope that upon inspection, you will acknowledge that although there is room for improvement in quality in future work (as the goal of this study is not to produce a finalized product but to advance research on generative models under raster supervision), **Grimoire demonstrates its potential to effectively generate vector graphics that closely resemble authentic SVG creations**.
>
> We believe this method can clearly be beneficial for practical use cases such as logo design.
>
>
>
> In the Showcase folder, we have also included a **font that uses letters and numbers generated by Grimoire**. We encourage you to install the .TTF file and try out the generations.

---

> > ### Author Response · Authors · 2024-12-03
> >
> > Dear Reviewer `6k4C`,
> >
> > This is a gentle reminder that the extended rebuttal period is nearing its close. We would like to inquire if our last response addresses your primary concerns about the practical use cases of our work.
> >
> > If it does, we kindly request that you reconsider the score.
> >
> > Thanks again for dedicating your time to enhancing our paper.

---

### Author Response · Authors · 2024-11-27

We sincerely thank the reviewers for taking the time to review our manuscript and providing valuable feedback.



As a summary, we are grateful to see that the reviewers found our approach of using raster image supervision for SVG generation novel [`Zk2S`, `ymmo`] and appreciated our detailed experiments, ablation studies and comparisons to baselines [`Zk2S`, `ymmo`, `sQs2`,`6k4C`]. Furthermore, reviewers acknowledged the excellent reproducibility of our work [`6k4C`, `Zk2S`].



We addressed the major concerns noted by the reviewers and provided the following enhancements in the revised version of our paper:



1.  **Improving the presentation of the paper** [`ymmo`]: We have carefully reviewed and addressed the suggestions from all reviewers, including adding a dedicated section for experimental settings, streamlining the description of the method, an additional diagram explaining the VSQ model, updating the label placements next to each equation, and providing additional clarification on the image encoder.

2.  **Comparison with SDS methods** [`Zk2S`]: We have included detailed comparisons with state-of-the-art SDS methods.

3.  **Additional detailed experiments**[`sQs2`]: We have conducted new experiments using segmentation-guided patch extraction, as suggested.

4.  **New qualitative results** [`sQs2`, `Zk2S`]: We added qualitative results on a more complex dataset (emoji) by extracting layers using a segmentation network. We hope that this will ultimately address the concerns regarding the complexity of the data.





With Grimoire, we introduce a primitive-agnostic scalable framework to generate SVG under raster supervision. Following this rebuttal, Grimoire now includes three distinct extraction pipelines, several compositing functions, and experimental results across four datasets, with the addition of preliminary experiments on emojis.

---

### Meta-Review · Area_Chair_sKva · 2024-12-22

**Metareview:**

The paper adopts the VQVAE framework for SVG generation using only image supervision by integrating a differentiable rasterizer. It received mixed/borderline ratings from the reviewers.

On the good side, the contribution of the work is recognized by most reviewers and the area chair. Applying the VQVAE framework to the field of SVG generation is an interesting and innovative approach. The use of visual guidance for supervision is convenient and eliminates the need for additional annotations, making the framework practical. The open-sourced code is a significant contribution to the community, enabling reproducibility and further exploration.

However, the writing has been a crucial concern for both Reviewer ymmo and Reviewer Zk2S. While some of Reviewer ymmo’s concerns were addressed, the overall clarity of the paper remains insufficient, especially the issues raised by Reviewer Zk2S. A specific issue lies in the inconsistent explanation of the mapping between patches and codes.

For example: On L213 of the revised paper: “The VSQ encoder maps each patch to $\xi$ codes on the hypercube,” suggesting that a patch corresponds to $\xi$ codes. On L262: “Each patch can be mapped onto an index code  of the codebook,” implying that a patch corresponds to a single code. This inconsistency is confusing, and the rebuttal failed to clarify it. Moreover, the revised appendix exacerbates the confusion: In Section A.3, it states, “In practice, for our experiments, we keep $\xi=1$ and disable the product of all codes that create the final embedding for the decoder.” If a patch corresponds to only one code, why mention $\xi$ in the first place? Does this imply some special design? How could a patch produce more than one code? The appendix also introduces the concept of “product of all codes,” mentioned in both L922 and Figure 15, which never appears in the main paper. This unexplained term further muddles the presentation.

**The lack of clarity in these fundamental aspects makes the paper difficult to understand and unsuitable for acceptance until the writing is substantially improved for better comprehension.**

Lastly, some other issue: Reviewer Zk2S noted that the produced shapes are overly simple, and the paper lacks results on more complex and diverse datasets. Limited technical novelty as raised by Reviewer 6k4C and Reviewer sQs2. While the rebuttal addressed Reviewer sQs2’s concerns and provided more complex results, Reviewer 6k4C remains unconvinced. Reviewer 6k4C also found the quality of the results less satisfactory, and the rebuttal does not convince Reviewer 6k4C on this issue.

**Additional Comments On Reviewer Discussion:**

The discussion highlighted several key issues with the paper, particularly the concerns raised by Reviewer Zk2S and Reviewer 6k4C. The rebuttal addressed some concerns but failed to resolve critical issues around the writing of the paper and the quality of results.

Reviewer Zk2S emphasized the poor clarity in explaining core concepts, such as the mapping of patches to codes and the role of $\xi$. While the rebuttal attempted to address these points, the explanations in both the revised main paper and the appendix introduced additional confusion. Concepts such as “product of all codes” were introduced without sufficient context, leaving significant gaps in understanding. This writing issue remains the most critical obstacle, as it affects the overall comprehensibility of the work.

Reviewer Zk2S also raised concerns about the simplicity of the produced shapes and the lack of results on more complex and diverse datasets. These concerns were addressed in the rebuttal. Reviewer 6k4C highlighted the limited technical novelty and the quality of results, which were not convincingly resolved in the rebuttal. While Reviewer sQs2’s concerns regarding novelty were addressed, Reviewer 6k4C remained unconvinced.

While the rebuttal provided additional details and addressed some of the reviewers’ feedback, the fundamental issues of writing clarity remain unresolved. The area chair also thoroughly read the paper and concurred with the reviewers on the fundamental writing issues.

---

### Decision · Program_Chairs · 2025-01-22

Reject